



# Evaluating a prediction system for snow management

Pirmin Philipp Ebner[1], Franziska Koch[2], Valentina Premier[3], Carlo Marin[3], Florian Hanzer[4,5], Carlo Maria Carmagnola[6,11], Hugues François[7], Daniel Günther[4], Fabiano Monti[8], Olivier Hargoaa[9], Ulrich Strasser[4], Samuel Morin[6], and Michael Lehning[1,10]

[1]WSL Institute for Snow and Avalanche Research SLF, Davos, Switzerland
[2]Institute of Hydrology and Water Management, University of Natural Resources and Life Sciences (BOKU), Vienna, Austria
[3]Institute for Earth Observation, EURAC, Bolzano, Italy
[4]Department of Geography, University of Innsbruck, Austria
[5]Wegener Center for Climate and Global Change, University of Graz, Austria
[6]University of Grenoble Alpes, Université de Toulouse, Météo-France, CNRS, CNRM, Centre d'Etudes de la Neige, 38000 Grenoble, France
[7]University of Grenoble Alpes, Irstea, LESSEM, Grenoble, France
[8]ALPsolut S. r. l., Livigno, Italy
[9]SnowSat, Grenoble, France
[10]School of Architecture, Civil and Environmental Engineering, École Polytechnique Fédérale de Lausanne, Lausanne, Switzerland
[11]Dianeige, Meylan, France

**Correspondence:** Michael Lehning (lehning@slf.ch) and Franziska Koch (franziska.koch@boku.ac.at)

**Abstract.** The evaluation of snowpack models capable of accounting for snow management in ski resorts is a major step towards acceptance of such models in supporting the daily decision-making process of snow production managers. In the frame of the EU H2020 project PROSNOW, a service to enable real-time optimisation of grooming and snow-making in ski resorts was developed. We applied snow management strategies integrated in the snowpack simulations of AMUNDSEN, Crocus

5   and SNOWPACK/Alpine3D for nine PROSNOW ski resorts located in the European Alps. We assessed the performance of the snow simulations for five winter seasons (2015-2020) using both, ground-based data (GNSS measured snow depth) and space-borne snow maps derived from Copernicus Sentinel-2. Particular attention has been devoted to characterize the spatial performance of the simulated piste snow management at a resolution of 10 meters. The simulated results showed a high overall accuracy of more than 80 % compared to the Sentinel-2 data. Moreover, the correlation to the ground observation data was high.

10  Potential sources for the overestimation of the snow depth by the simulations are mainly due to the impact of snow redistribution by skiers or spontaneous local adaptions of the snow management, which were not reflected in the simulations. Subdividing each individual ski resort in differently-sized ski resort reference units (SRU) based on topography showed a slight decrease in mean deviation. Although this work shows plausible and robust results on the ski-slope scale by all three snowpack models, the accuracy of the results is mainly dependent on the detailed representation of the real-world snow management practices in

15  the models. This calls for an assessment of impacts from meteorological station measurements and their interpolations in the ski resorts as well as potential limitations in describing the snow cover, especially managed snow, by simulations.





# 1   Introduction

The Alpine ski industry plays a central economic role in many mountain regions and is important for regional development.
About 13.6 million people live in the European Alpine region with around 60 to 80 million tourists visiting every year. The ski
resorts generate a high turnover in the winter tourism destinations (Vanat, 2020). However, a multi-annual perspective shows a
stagnation of skier visits and the growing economic pressure on resorts, aggravated by the threat of climate change and decrease
in average snow conditions as well as the consequences of the Covid-19 pandemic since February 2020. The trend to early
winter demand for perfect ski slopes is still increasing but climate change renders a reliable early ski slope preparation more
and more challenging. This was particularly visible throughout most of the European Alps at the beginning of the snow seasons
2014/2015 and 2015/2016, with strong deficits in natural snow amounts and elevated temperature conditions (NOAA, 2014,
2015) hampering the possibility to produce machine made snow. Ski resorts throughout the world have become increasingly
reliant on snow-making facilities to complement the natural snow cover. Over 90 percent of all large ski areas in the Alpine
region use snow-making facilities. In terms of snow-making, Italy (90 %) is in the lead followed by Austria (70 %), Switzerland
(45 %), France (35 %) and Germany (25 %) (Lalli et al., 2019). The whole workflow of ski resort management with modern
slope preparation and maintenance, snow storage and machine made snow, however, could increase in efficiency. Examples of
potential savings are related to both i) an over-production, which leads to a delayed melt out in spring well after the closing
of the season and ii) a too early snow making production in autumn when the conditions can rapidly change and lead to the
complete melt of the produced snow, e.g. due to an early-season warm spell.

Beyond the time scale of weather forecasts, which are generally reliable for a time frame of a few days, ski resort managers
have to rely on various and scattered sources of information, hampering their ability to cope with highly variable meteorological
conditions. In the frame of the funded European Union's Horizon 2020 project PROSNOW (Morin et al., 2018), a demonstrator
of a meteorological and climate prediction and snow management system from a short-term forecast covering the first 4 days
to several months ahead, specifically tailored to the needs of the ski industry, was developed. Besides and within this project,
approaches for representing snow management in numerical snow cover models were developed (Hanzer et al., 2020; Spandre
et al., 2016) and integrated in the existing snow cover models AMUNDSEN (Strasser, 2008; Strasser et al., 2011; Hanzer et al.,
2016) SNOWPACK/Alpine3D (Lehning et al., 2006; Bartelt and Lehning, 2002) and Crocus (Vionnet et al., 2012; Lafaysse
et al., 2017). Applying these snowpack models capable of representing the effects of snow management (grooming and snow-
making), snow depth on the slopes could be predicted in real time and short-term and seasonal forecast mode for various ski
resorts.

To increase the adaptation capacity of the skiing industry, there is a great need to combine weather and climate forecasting,
snow modelling and observations, and to promote existing products to demonstrate their value for professional decision making.
In this context, in-situ observations as well as optical and microwave remote sensing have proven to be mature technologies
(Sirguey et al., 2009; Dumont et al., 2012; Bühler et al., 2015). For example, in terms of snow coverage, remotely sensed data





can provide twofold key information to the models. On one side, they can be used to correctly initialize the state variables of the models (Monti et al., 2016). On the other side, they can be used to cross-compare the results of the simulations and evaluate them (Mary et al., 2013; Notarnicola et al., 2013a, b). Since remotely sensed data mainly provide binary information on snow presence or absence, a complete model evaluation needs an additional comparison with measured in-situ observations (Hanzer et al., 2016). Nowadays it is increasingly common to employ advanced snow depth monitoring systems such as

Global Navigation Satellite System (GNSS) equipped grooming machines to track the snow depth on the ski slopes. Thereby, the performances of the models can be evaluated by comparing model outputs with the measured snow depth values. This evaluation strategy has been shown by Hanzer et al. (2020) for a single point on a slope for various ski resorts. In this work, the performance of the snowpack models were cross-compared with several snow depth measurements spatially distributed with a resolution of 10 m over different ski slopes for two winter season i.e., 2016/17 (driest winter in the recent years) and 2017/18

(snow-rich winter throughout the entire Alps) .

The objective of this paper is to evaluate the accuracy of the piste snow management module refined and implemented in the framework of the H2020 PROSNOW project embedded within several snowpack models to simulate snow managements in general and for each individual PROSNOW pilot ski resort in high spatial resolution. In a first step, the results of the snowpack simulations were compared both with remotely sensed satellite snow-covered maps and with snow depth measurements

spatially distributed along the ski slopes acquired with specific GNSS systems. The impact of elevation, slope and aspect as well as temporal aspects within a season or amongst different years were evaluated. In the second step, the simulation domain was spatially discretized into defined ski resort reference unit (SRU) (Hanzer et al., 2020) to reduce computational effort but still be able to achieve meaningful results. In this case, each ski resort was divided into a number of elevation bands ranging from 50 to 400 m.

## 2   Study sites, models and data

### 2.1   Pilot ski resorts

Within the PROSNOW project we focused on the following nine ski resorts, which are also all part of this study: Seefeld (cross-country part) and Obergurgl in Austria, La Plagne and Les Saisies in France, Garmisch Classic in Germany, Colfosco, San Vigilio, and Livigno in Italy, and Arosa-Lenzerheide in Switzerland. This selection of ski resorts represents a large diversity

of geographical, climatical and snow-making practices and equipment. Figure 1 and Table 1 show the locations and key characteristics of the pilot resorts.

### 2.2   Snowpack models

Snowpack simulations are performed with AMUNDSEN for the Austrian and the Italian resorts (Colfosco, Obergurgl, San Vigilio, and Seefeld), with Crocus for the French resorts (La Plagne and Les Saisies), and with SNOWPACK/Alpine3D for the

remaining resorts in Switzerland, Germany, and Italy (Arosa-Lenzerheide, Garmisch Classic, and Livigno). We used for each





ski resort different settings for the parameters concerning wet-bulb temperature, snow depth threshold, timing and density of grooming. A detailed description of the functionality and parameters of the snow-making and grooming modules, which are used in all of the snowpack models are shown in the study by Hanzer et al. (2020) and the applied configuration for each ski resort in Table 1. The three snowpack models, AMUNDSEN, Crocus, SNOWPACK/Alpine3D, are well-established and have

been widely applied in numerous studies throughout the past decades (Essery et al., 2020; Krinner et al., 2018).

All three models require spatial input data for the snow management simulations consisting of a digital elevation model (DEM) covering the study sites, the locations of the ski slopes, and the locations and types of the snow guns (lances or fans). The used snow management configurations for each ski resort are shown in Table 1. Meteorological forcing data for the simulations is based on measurements from automatic weather stations in close to or within the study sites and from the

SAFRAN analysis for Crocus model runs (Vernay et al., 2019), consisting of at least hourly measurements of air temperature, precipitation, relative humidity, wind speed, and radiation. The generated output data are rasterized snow depth files with a resolution of 10 m.

The models are equipped with a machine made snow production and grooming module, which can be used for the operational applications. A set of core parameters can be used for very detailed simulations of snow management practices in single ski

resorts. They take into account snow demand, the meteorological conditions including information on wet-bulb temperature and wind speed, and the ski resort infrastructure in terms of the amount of snow that can be produced in a given time step at a certain location within the resort. For the simulations it is assumed that for a given snow gun all of the produced snow is distributed immediately and evenly over a predefined slope section. Additionally, the grooming module allows to account for the distinct properties of groomed snow on ski slopes depending on the amount of snow present and a defined grooming

schedule. It assumes that grooming has no effect on the distribution of snow, e.g. shifting of snow from one place to another, but rather only compacts it (Hanzer et al., 2020).

## 2.3  Ski resort reference unit - SRU

Real time simulations with a very fine spatial resolution (i.e. 10 m) require a very high computational demand and such fine spatial resolutions are also often not necessary for the overall day-to-day resort management. Spatial clustering of slopes and

slope sections is often sufficient for the snow managers working in an operational mode. Therefore, we additionally discretized each ski resort in ski resort reference units (SRUs). In a post-processing step, we aggregated the initial 10 m pixel size to larger areas. We defined different SRU sizes of the individual pistes by slicing them into the following elevation step ranges: 50 m, 100 m, 200 m, 300 m and 400 m. This aggregation results in different amounts of SRUs for each considered elevation range. Figure 2 shows exemplarily a map for the western part of Arosa-Lenzerheide with the resulting elevation classes for the

different SRU sizes. In addition, we evaluated the sensitivity of SRU size to the final accuracy. More details about the SRU definition can be found in Supplementary Material A1 and Table B1.



## 2.4 Sentinel-2 data

The model results for all ski resorts were compared with remote sensing images; for this study, Sentinel-2 data (S2) were used. The processing of the S2 snow-covered maps was done in three main stages: i) calibration to Top of the Atmosphere

(ToA) reflectances, ii) re-projection, resampling and co-registration with the model grid with a final resolution of 10 m, iii) classification with a Support Vector Machine (SVM) classifier trained with an active learning procedure (Tuia et al., 2016). In particular, the classification was devoted to both: i) detecting the clouds; and ii) detecting the snow presence. This was done by exploiting the most representative features for each of the two classification problems. In detail, we used all the spectral bands, the normalized difference snow index (NDSI) and the normalized difference vegetation index (NDVI) for the

cloud classification. NDVI was calculated as normalized difference between NIR and red bands, whereas NDSI was calculated between green and SWIR bands. Regarding the snow detection, we used the spectral bands, NDSI, NDVI and the illumination angle calculated from the solar zenith and the solar azimuth angle (Riaño et al., 2003).

Since the S2 snow maps are compared to the snow simulations, particular attention was devoted to obtain accurate results. For this purpose, three main steps were performed to address the main problems related to snow classification from optical

images, which are the detection of: i) particular cloud conditions; ii) the snow mixed pixels i.e., pixel in which other classes than the snow contribute to the observed spectral response; and iii) the snow under the canopy of the forests.

First, we performed a visual analysis of all the S2 images for excluding the scenes presenting complex cloud conditions. In particular, semitransparent clouds, which are thin, high-altitude clouds composed of ice crystals were detected from the S2 band acquired at 1.375 $\mu$m. Interestingly, semitransparent clouds might not be visible in other spectral bands but they alter the

spectral signatures leading to unreliable results.

The SVM classifier was trained in a way that a pixel with at least 50 % of snow coverage is classified as snow. This means that for example during the snow making production at the beginning of the season a pixel in which a significant snow production is ongoing is classified as snow even though not all the pixel area is covered with snow. Additionally, we identify the shadowed areas from where the multi spectral sensor on board of S2 is not able to record sufficient energy for distinguishing between

snow and snow free areas. This is happening when the sun is low at the horizon approximately from mid November to mid February and the terrain is extremely steep. In all the other shadow cases, the SVM classifier was trained to detect the snow presence. Hence, the output of the procedure was a classified map with four classes, i.e. snow, snow-free, shadows and clouds.

Since the detection of snow under forest canopy is a challenging research topic from both the remote sensing and modeling point of view, we conservatively masked out forested areas for all the ski resorts, based on the landcover classification provided

by OpenStreetMap (OSM) (resolution of 30 m) (Schultz et al., 2017) rasterized to a resolution of 10 m and manually refined for all ski resorts in such a way that the ski slopes that were passing through forest but were visible at the resolution of S2 were considered for the evaluation. The masked layers include coniferous, deciduous and mixed forests and in some particular cases also scrubs and heath. After this screening all the snow maps were considered reliable and the scenes with a percentage lower than 50 % of cloud were retained as sufficient information for the cross-comparison.





A detailed overview of the number of available S2 scenes for each year and ski resort is presented in Table A1 and Figures A1 and B1 in the Supplementary Material. The number of S2 scenes which were available for each ski resort within the winter periods of 2015 to 2020, is in general high and ranged between 62 and 190 per ski resort. The differences in numbers of available scenes was mainly affected by cloud coverage and the atmospheric condition to perform accurate classification.

## 2.5    GNSS snow depth data

More and more ski resorts are relying on spatially distributed snow depth measurements performed with a modern Global Navigation Satellite System (GNSS) technology for an efficient management of their slopes. This technique relies on differential GNSS signals and takes measurements without snow depth on the slopes as a reference into account. The sensors are installed on top of the groomers and thereafter snow depth can be tracked as a positive side effect whilst grooming the pistes. This technology ensures a snow depth measurement accuracy down to the centimetre level and in a spatial resolution of 1 m, which

allows also to track snow redistributions with the groomers.

For our study, rasterized data were provided by Snowsat and Leica and were resampled to a resolution of 10 m using the average value in order to be directly comparable with the model outputs. The GNSS snow depth data were available for almost all pilot ski resorts including Arosa-Lenzerheide, Garmisch Classic, Livigno, Obergurgl, Seefeld, San Vigilio, Colfosco and Les Saisies. We considered for the analysis the measurements spanning from the 1st of December to the 31st of March with a

daily temporal resolution, when GNSS data were available. The data have been preprocessed to eliminate outliers and to check their consistency. Table 1 shows the available seasons of GNSS snow-depth measurements for all ski resorts.

## 3    Evaluation

In this section, we describe how we evaluate the snowpack simulations carried out for the PROSNOW ski resorts. This includes i) the evaluation of simulated snow depth and ii) the evaluation of snow-covered area. In detail, the simulated snow depth was

compared with the GNSS-derived measurements over a number of ski slopes, whereas the snow-covered area is evaluated by comparing the model snow-covered area with the S2 snow maps. The metrics used for assessing the agreement between the simulations and S2 snow maps are the confusion matrix and the snow persistence index defined below.

The evaluation analysis for both snow depth and snow-covered area was conducted by stratifying the data according to temporal and topographical constrains. Moreover, a differentiation was made between natural snow i.e., snow outside the

pistes and managed snow i.e., snow inside the pistes. In the following subsections more details on how the evaluation metrics were calculated will be presented.

### 3.1    Snow coverage - Snow persistence (SP) and Confusion matrix (CM)

The latitude and elevation of a ski resort has a big impact on the timing of snow accumulation and melt. Therefore, comparing snow patterns between regions is challenging despite the widespread application of remotely sensed methods for snow research.

The snow persistence (SP) is a snow metric that can be used to map snow zones globally (Macander et al., 2015; Wayand





et al., 2018; Vionnet et al., 2020). It is calculated in this work as the number of snow-covered days divided by the number of valid Sentinel-2 observations for the whole period (5 years). The number of total observations can vary for each pixel since cloudy/masked pixels are not considered for the computation. Hence, the same valid dates are considered for the model. The resulting SP Index ranges from 0 to 1. This approach has been used in a wide range of climates (Richer et al., 2013; Moore
et al., 2015; Saavedra et al., 2017), to identify transitions between rainfall and snow melt peak streamflow source regimes (Kampf and Lefsky, 2016) and for predicting water yield (Saavedra et al., 2017; Hammond et al., 2018).

Two SP indices were extracted considering both S2 snow maps and model simulation. They were calculated pixelwise as the ratio between the number of snow-covered days derived by S2 or from the model, divided by the total number of S2 observations (snow or snow-free). The values of SP were always between 0, i.e. always snow-free dates, and 1, i.e. always snow-
covered dates. If a S2 snow map pixel is classified as cloud the corresponding snow model output is masked out preserving the one to one correspondence between the two SP indices.

Additionally to the SP index, for each ski resort a confusion matrix (CM) was computed to assess the quality of the S2 and snowpack simulations. We refer the confusion matrix to modelled vs. observed variables. The confusion matrix has the form indicated in Table 2: TP: true positive, i.e. both model and S2 labelled as snow; FP: false positive, i.e. model labelled as snow,
S2 as snow free; FN: false negative, i.e. model labelled as snow free, S2 as snow; and TN: true negative, i.e. both model and S2 labelled as snow free. Therefore, TP and TN indicate that model and S2 data match with each other whereas FP and FN indicates that they don't match.

We distinguish between natural snow and snow on the slopes. Furthermore, the analysis was split in three periods: beginning (B: October-November-December), middle (M: January-February) and end (E: March-April-May) of the season. A pixel can
be either true (snow in S2 data and model; no snow in S2 data and model) or false (snow in S2 data and no snow in model; snow in model and no snow in S2 data). With the accumulation of all pixels assigned to be true, an overall agreement OA (%) was calculated for each period and catchment:

$$OA = \frac{TP + TN}{TP + FP + TN + FN} \tag{1}$$

which describes how often the agreement between S2 and the simulations was correct.

## 3.2  Snow depth - Root mean squared deviation (RMSD) and mean deviation (MD)

The metric used for this assessment are the mean deviation (MD) and the root mean square deviation (RMSD) over time for each ski resort. Regarding the snow-covered area evaluation, the binarization of the simulated snow depth to snow-covered map was done by imposing a threshold of 0.05 m i.e., every value above this threshold was identified as snow, while on the contrary all the values below 0.05 m were identified as snow free. This threshold is in line with previous works (Notarnicola,
2020). In each defined SRU we analyzed the variations in terms of snow depth MD and RMSD. The MD was used to relate between the modelled and measured snow depth on the slopes among years for the period October to May. Measured snow depth data from all available years of observations (see Table 1) were used in order to analyse the quality of the simulated snow management configuration for each ski resort. This allowed better projection of the uncertainty of the simulated snow





management configuration based upon the measured snow depth. RMSD and MD per one time point are defined as:

$$\text{RMSD} = \sqrt{\frac{1}{N}\sum_{i=0}^{N}(SD_{\mathrm{m}}(i) - SD_{\mathrm{s}}(i))^2} \qquad\qquad \text{MD} = \frac{1}{N}\sum_{i=0}^{N}(SD_{\mathrm{m}}(i) - SD_{\mathrm{s}}(i)) \tag{2}$$

where $N$ is the number of valid pixels for a given date, $SD_{\mathrm{m}}$ is the GNSS snow depth measured and $SD_{\mathrm{s}}$ is the snow depth simulated by the model. Hence, the metrics were calculated pixel based and only on those pixels where GNSS measurements were present, i.e. the number of considered pixels $N$ can vary for each date. A negative (positive) MD value indicates an overestimation (underestimation) of the snow persistence of the PROSNOW models.

## 4 Results

In this section, the simulated results for all ski resorts were compared with the S2 and with the GNSS measured snow depth data on the ski slopes. Model runs were performed for five winter seasons (2015-2020) from 1st of October until end of May. The simulations were carried out for all ski resorts using the default snow management configurations accounting for both fan guns and lance guns as well as different temperature threshold and base-layer production targets, given in Table 1. The configurations were either assumed or provided by the responsible person of snow making of each ski resort.

### 4.1 Snow coverage

The S2 algorithm produced accurate snow maps with an overall accuracy above 80 %, both for high-alpine and lower-lying mountainous regions, and different stages of the season like season start, mid season and end of season. A first approach to assess the skills of the models using S2 maps as a reference was a confusion matrix. Table 3 presents the results for each ski resort. Analyses were carried out for solely regarding snow on the pistes and additionally also for the complete ski resorts including the off-piste areas. Moreover, overall accuracy trends over time are shown for each ski resort in the supplementary material (Figures B1 and A1). The overall accuracy was over 80 % and the accuracy was highest in the middle of the season, with almost 90 % or higher. The snow distribution was well simulated at the beginning and end of the season, with an agreement of over 79 %. However, a larger mismatch was observed for San Vigilio with 69 % at the end of the season. A higher agreement between simulation and the S2 data was observed when regarding the pistes only. Compared to the analysis of the entire resort including natural snow, the overall accuracy of pistes only was up to 8 % higher. Only one resort (Garmisch Classic) showed a slightly lower accuracy of 8 % on pistes.

The snow coverage quality of the snowpack simulations using the snow management modules was further assessed using the S2 data and SP indices. The SP indices were calculated for the simulations and S2 data and the relative differences are shown in Figure 3. The green pixels indicated the applied forest mask to minimize the underestimation of snow detection by the S2 algorithm. Observed SP presented similar patterns for each ski resort showing that snow persistence patterns were primarily controlled by the elevation. High elevation corresponds to high SP values, whereas low SP values were found near the tree line and lower elevations. In addition, low SP values were also found near ridge lines, exposed to wind, and influenced by lateral





snow redistribution and snow accumulation. These effects were not captured by the simulations. Further, the derived SP values
were dependent upon the elevation and slope orientation, primarily due to the impact of solar radiation on simulated snow
ablation. The biggest difference in the SP index between the S2 data and model was found in steep slopes where the effect of
snow gliding (e.g. avalanches) is stronger.

The overall accuracy was mainly impacted by the elevation and slope. An interesting analysis was represented by the trend
of the overall accuracy over 100 m elevation classes, 5 degrees slope - and 45 degrees aspect classes (i.e., North, North-East,
East, South-East, South, South-West, West, North-West). This analysis was carried out taking specifically only the snow on
the pistes into account and is presented in Figure 4 (left). The agreement of the simulations with the S2 data increased with
increasing elevation by around 1 % per 100 m elevation class. The overall accuracy over 5 degrees slope classes is not affected
by the slope steepness and the orientation of the slope towards the sun showed no influence on the accuracy.

A closer look at the SP index showed an elevation dependency, but a clear slope or aspect dependency is hard to detect.
Figure 4 (middle) shows the difference between S2 and model SP index, similar to Figure 3. Positive values indicated an
underestimation of the model respect to S2. Above 2000 m, the MD SP index is close to zero and the model results are
consistent with the S2 data. However, below 2000 m the MD increases as most of the slopes are equipped with snow-making
facilities and are affected by local snow management adjustments due to local snow and weather conditions.

## 4.2  Snow depth

The simulated default snow management configurations reproduced the actual conditions in all ski resorts well. Figure 5 shows
the RMSD and the MD between the modelled and GNSS measured snow depth over time for each ski resort and each season.
Especially at the beginning of the season, the RMSD values between the simulated and the GNSS-measured snow depth on
the pistes was below 0.4 meters on average. However, the RMSD values slightly increased during the season affected by
daily adaption of the snow management configurations due to the actual weather conditions. The complexity of the snow
management configuration increased during the season leading to an increase in the RMSD values at the end of the season. In
general, the temporal evolution of the RMSD values shows almost no large peaks. The large peak at Lenzerheide on January
15th, 2018/2019, occurred due to a heavy snowfall period which led to an extraordinary avalanche situation. As a result, a large
part of the ski area was closed and many slopes were no longer groomed. This led to the strong increase in the RMSD values
in this season. In general, our models mainly overestimated the snow depth and the fluctuations increased during the season.
Regarding the degree of linear dependency of the simulated and measured data, the MD as shown in Figure 5 reaches in some
resorts more than 0.5 meters suggesting that more snow was produced compared to the model. For several resorts an increase
in MD can be observed especially at the end of the seasons.

Figure 4 (right) represents the MD trends between the simulated and GNSS snow depth data for 100 m altitudinal classes,
5 degrees slope classes and 45 degrees aspect classes. For this analysis, we considered RMSD values, which refer to all the
pixels corresponding to a fixed elevation class, degree slope or degree aspect class, respectively, calculated over all the available
days. Overall, the MD values vary between -0.6 and 0.6 m. No systematic relationship can be seen between the elevation and
steepness of the pistes, the curves are highly specific for each resort. However, regarding the elevation discretization, the MD





values increase on average with increasing altitude. An overall average MD trend on the 5 degrees slope classes is not found. Only for Arosa-Lenzerheide, Garmisch Classic and Les Saisies a decrease in MD values for steeper slopes was observed. The

analysis of the 45 degrees slope aspect bands on the MD values show that the accuracy decreases for slopes oriented mainly towards to the south-west. This is primarily due to the impact of solar radiation on the snow ablation.

### 4.3 SRU discretization

Discretizing the ski resorts in coarser clusters allows minimizing the error in terms of RMSD due to averaging effects between the simulations and GNSS measured snow depth between 8 and 45 % independent of the season and cluster size (see Table

B1). Therefore, it was not possible to find an overall optimal SRU size regarding the RMSD values. We also computed the MD of the errors calculated as difference between the measured snow depth and the modelled snow depth, shown in Figure 7. The original pixel snow depth is kept for the 10 m resolution, while a mean snow depth for each SRU is calculated for the different SRU altitudinal band classes. For the computation of the SRU snow depth, only those pixels with valid GNSS measurements were considered and taken into account (between 0 and 3.5 m). The MD and the standard deviation of the error

are then calculated by considering all the available measurements over time and space. We propose here a differentiation for the four months December, January, February and March in Figure 7. Interestingly, the mean and the deviation of the MD values differ for single resorts widely.

In addition, we tested the spatial variability within the pistes. For a visualization example presented in Figure 6 we chose a long piste ranging between 700 and 1700 m a.s.l. within the Garmisch Classic ski resort for a date with maximal piste

coverage. Figure 6 reports the simulated snow depth and the GNSS measurements as well as the pixelwise difference, for the original resolution at 10 m (on the left). The same variables averaged over a possible SRU discretization, in this specific case considering 100 m elevation bands, are also reported in the same figure on the right. Regarding the fine resolution of 10 m, it is obvious that the GNSS snow depth data is much more variable in general and as especially occurring in the upper part of this piste due to highly localized snow management than the simulations with the default configuration can provide. A good

agreement was found especially in the lower and middle parts of this example slope with slight under and over estimations of the simulated snow height. The picture would look similar for all other pistes and resorts (not shown).

## 5 Discussion

This study presents a new high-resolution evaluation of snowpack simulations including snow management modules for mountain ski resorts to assess the quality of the simulations. The simulated results showed a high overall accuracy of more than 80 %

compared to the Sentinel-2 data and a root mean squared error to the GNSS measured snow depth below 0.6 m. The simulated results for all ski resorts are plausible and robust on the ski slope scale.



## 5.1 Snow coverage

For every ski resort, a large number of S2 images classified with high accuracy were available to assess the quality of the simulations in terms of snow coverage. More than 62 S2 images at each ski resort and even more than 150 for two resorts were analysed. The specific machine learning algorithm to derive information with low uncertainty about the presence/absence of snow from the Copernicus Sentinel-2 images allow the generation of snow maps across the Alps with a relative low manual effort. Additionally, the very detailed forest mask applied to the evaluation allowed us at the same time i) to avoid situations for which the information provided by S2 are insufficient to produce accurate results; and ii) to be able to extract information about the snow cover for the pistes crossing dense forests such as it is often the case for Garmisch. This allowed us to minimize the pixels loss due to canopy shading. The highest pixel losses with respect to the total pistes area were in Arosa-Lenzerheide (5.7 %) and Seefeld (5.5 %), followed by Obergurgl (2.4 %), Livigno (0.2 %), and Colfosco (0.2 %). For the other ski resorts it was zero.

The underestimation in the overall accuracy at the beginning and the end of the seasons was due to the fact that sometimes the exact snowline in the S2-data was hard to detect. Some snowlines were obscured by shadows and they often did not appear as continuous lines, which may slightly bias our regional estimates especially in areas which span different land use. Also white rock types or illuminated wet rocks can lead to brighter pixels in the S2 data and to an underestimation in the overall accuracy. Due to the similar spectral signal of these pixels to snow, the algorithm detects an ice-snow boundary and classifies these pixels as part of the snowlines. Since these patches were situated in ski resorts in rocky areas, these misclassified pixels introduced negative biases in the overall accuracy estimates, and they were filtered out manually.

By inter-comparing the model simulations and the S2 images we encountered some recurrent errors. As described in the previous section, in average the accuracy of the simulations are high and the snow coverage simulated by the PROSNOW models are consistent with observations. However, wrong discretization and/or missing meteorological input and lack of snow managing/land use information were the main sources of errors. In particular, (1) ephemeral snow (i.e., snow that lasts few days either at the beginning or at the end of the season) are difficult to be simulated correctly by the models; (2) rain/snow transition e.g., rapid snow melt inside the catchment are hardly to be matched correctly by the models; (3) due to unknown snow making strategies, which are then not incorporated in the PROSNOW models, snow making at the beginning of the season and delayed/anticipated snow melting at the end of the seasons are not correctly modelled over managed slopes; (4) the heterogeneous landscape at 10 m resolution plays a role in the snow accumulation and melting dynamics e.g., towns, lakes and roads are visible and change the snow distribution. This information is generally not addressed by the PROSNOW models. These are just confirmations of the expected limitations of the state of the art snow cover models. However, for the first time a systematic and extensive evaluation at high model resolution was performed. The details of the analysis with all the different recurrent errors encountered for each ski resort is shown in Table C1 and can be used for future studies. Note that the simulations presented here do not benefit from snow depth measurements and water consumption of snow-making, which are used, upon availability of the relevant data, for real-time applications of the models for operation PROSNOW service provision, thereby limiting the impacts of some of the caveats identified above.





## 5.2 Snow depth

Using the extracted GNSS data which was derived by grooming the pistes on a daily mode as ground observation can be only accepted with some restrictions. There are several problems which might affect the quality of the data: (i) the digital elevation model (DEM) profile of all ski resorts is prone to change every year due to earthwork and adaption of the slopes and

is always considered correctly in the extraction of the snow depth; (ii) the inclination of the groomer has a large impact on the GNSS measured snow depth. For example if the calculated snow depth is 0.5 m, the effect of 30 % inclination would be 6 cm which means that the calculated snow depth would be 12 % higher than in reality. Furthermore, the work of the groomers is not only to measure the local snow depth but also to fill sinks, compensate humps or level out the snow production or redistribution by the skiers. This might lead to very small scale variances in the GNSS-derived snow depth both for pistes with

and without technical snow production, whereas the model shows less pronounced variability. Therefore, especially the small-scale deviations between the simulated and GNSS-measured snow depth are not well applicable to reflect the model accuracy. The comparison rather shows to which degree the default snow management configuration was applicable for the individual ski resorts. As each ski resort spontaneously adapts its snow management production due to changing weather and snow height conditions, it is not possible to consider the spontaneous snow management adjustments during the season in the simulations. It

does not make a big difference between piste with and without technical snow production despite the models do not explicitly simulate the snow redistribution. However, the RMSD between the modelled snow depth and GNSS measurements over time for the ski resorts are in the uncertainty range of the simulated configuration. Hanzer et al. (2020) analyzed ten different snow management configuration and an uncertainty range in snow depth of 20-50 cm was observed. This shows that the default snow management configuration leads, in general, to satisfactory simulation results for the individual ski resorts.

Some recurrent errors are encountered by inter-comparing the model simulations and the GNSS measured snow depth. The quality of the simulations is high and they showed plausible and robust results on the ski slope scale. However, there are still sources of errors: (1) extreme snowfall situations; (2) snow redistribution (i.e south, south-west exposed pistes) are not considered in the simulations; (3) levering of the pistes to reduce the accident risk of the skiers; (4) snow redistribution by the skiers; and (5) systematic errors due to wrong snow making strategy. These errors are already known but it is too complex

and not straightforward to consider this in the simulations at current state. In detail, slightly different recurrent errors are encountered for each ski resort by inter-comparing the model simulations with GNSS measured snow depth, which is shown in Table C1.

## 5.3 SRU discretization

Scale and data aggregation have important effects on the simulations and interpretation of snow depth data, which is also true

for snow on pistes. Studies in different research fields clearly demonstrate that spatial variability and statistics are dependent on scale (Marceau, 1999; Wu, 1999; Wu et al., 2000; Perveen and James, 2010). Our evaluation showed that these scale dependencies in spatial variability and statistics appear complex with non-linear increases with increasing grid-cell size. However a simple relationship between variability and scale emerges upon closer inspection: Aggregating the 10 m simulated pixels to





different SRUs sizes led to a decrease of the error compared to the GNSS measurements due to averaging effects of the high
spatial variability of the GNSS snow depth data. The RMSD values decreased between 8 to 45 % regarding the SRU size using
an altitudinal band of 50 m, however this effect diminished by further coarsening.

Nevertheless, the identification of guiding principles for researchers to combine data and models at different spatial and
temporal scales and to extrapolate information between scales still remains a challenge. By going from fine to coarse scales,
aggregation and generalization set in. The rate of information loss is influenced by small-scale spatial snow production and
grooming patterns. Heterogeneous snow production for instance lead to more information loss than aggregations at coarser
scales. The spatial small-scale effects of moving snow by the groomers or skiers for instance, disappears slowly with decreasing
resolution and those that are dispersed are lost rapidly. This leads to an under- or overestimation of the simulated snow height.
Therefore, a methodology needs to be developed to find out how much the loss of information takes place. Multiscale analysis
was necessary to show that variability for different aggregation types of the snow height is inherently different, and that for
each SRU this can be different as shown in Figure 4.

However, we demonstrated a consistent evaluation procedure for all the PROSNOW ski resorts that can be useful for snow-
pack modelers and ski resort managers. An understanding of the nature of scaling effects is needed, when spatial or temporal
scale is an independent variable. In landscapes with homogeneous snow depths, where snow measurements can be summed
directly, such scale problems may not occur. However, in snow distributed landscapes like in ski resorts, snow measurements
obtained at fine scales often cannot be summed directly to produce regional estimates. Therefore, reasonable measures are not
always given by weighted averages because heterogeneity in snow production and distribution may influence scaling processes
in nonlinear ways. In such cases increasing the level of spatial heterogeneity may also increase the difficulty of extrapolating
information across scales (Perveen and James, 2010).

### 5.4 Remote sensing and GNSS snow depth

The use of remote sensing and GNSS data allows to define a evaluation procedure for the snowpack models and helps to
improve the resource management of the ski resorts. The GNSS data provides means to collect accurate snow depth points in
the field for precise correction of the simulated snow depths. Further, using snow depth measurements over an entire season
together with snowpack simulations is a powerful tool in the long term. It allows to estimate the minimum snow depth required
at various slope sections. This ensures that slopes are optimally prepared and groomed right up to the end of the season. The
spatial remote sensing images are needed to improve the simulated snow covered area especially at the beginning or end of
the season, or for lower situated ski resorts where natural snow precipitation is low. Further, it allows to correct the simulated
ablation process at the end of the season and the snow in process at the beginning of the season.

The combination of both techniques allows to evaluate and initialize the simulations: imagery can be used for primary
digitization of the snow cover where GNSS can be used for in-situ observation of the snow height for the simulations. A
detailed analysis of the differences between the two methods will allow us to make better decisions about when and how much
snow is distributed by both groomers and skiers. Further, the effect of snow melting and snow redistribution by wind on the

slopes can be extracted. This allows us to improve the models. However, this is not easy to extract and was not within the scope of this paper but should be considered for further studies.

# 6    Conclusions and Outlook

The initiative for this study emerged within the H2020 PROSNOW project to evaluate the snow simulations over the nine PROSNOW pilot ski resorts by comparing model outputs with local and remotely-sensed measurements in terms of snow coverage, persistence and snow depth. The three snowpack models AMUNDSEN, Crocus and SNOWPACK/Alpine3D include all piste management modules and were evaluated using both, ground-based data (GNSS measured snow depth) and spaceborne Copernicus Sentinel-2 derived snow maps (i.e., machine learning was exploited to derive information with low uncertainty

about presence/absence of snow). The evaluations were performed for five winter seasons (2015-2020) from 1st of October until end of May and have been performed in a stratified manner in order to assess the performance of the snow simulations under different conditions. Particular attention has been devoted to characterize the spatial performance of the snowpack models with integrated snow management moduls. Our presented results show high accuracy of the simulations representing the 'reality' well.

An inter-comparison of the three snowpack models applied to the same resort would be a logical next step from the model development perspective. Differences in the simulated results of the three models for a given ski resort would be mainly due to the different implementations of the snow management configurations into each model and due to the different snowpack energy and mass balance approaches. Such effects should have to be disentangled and discussed accordingly to have a reasonable comparison. However, this is not straightforward and is out of scope for this paper, but should be considered for the future.

Nevertheless, this work showed that all three snowpack models applied for piste management reproduced plausible and robust results on the ski slope scale and the overall accuracy of the results is mainly dependent on the degree to which the real-world snow management practices are integrated.

*Code and data availability.* Datasets related to this article can be found at https://doi.org/10.5281/zenodo.4541353 or asked directly by the authors.

*Author contributions.* Pirmin Philipp Ebner: Conceptualization, Methodology, Software, Resources, Validation, Visualization, Writing - Original Draft

Franziska Koch: Conceptualization, Methodology, Software, Resources, Writing - Reviewing and Editing

Valentina Premier: Conceptualization, Methodology, Software, Resources, Writing - Reviewing and Editing

Carlo Marin: Conceptualization, Methodology, Software, Resources, Writing - Reviewing and Editing

Florian Hanzer: Software, Resources, Reviewing

Carlo Maria Carmagnola: Software, Resources, Reviewing



Hugues François: Software Resources, Reviewing

Florian Hanzer: Software, Resources, Reviewing

Fabiano Monti: Resources, Reviewing

Olivier Hargoaa: Resources

Ulrich Strasser: Supervision, Reviewing and Editing

Samuel Morin: Supervision, Reviewing and Editing, Funding acquisition

Michael Lehning: Supervision, Conceptualization, Methodology, Writing - Reviewing and Editing

*Competing interests.* The authors declare that they have no known competing financial interests or personal relationships that could have

appeared to influence the work reported in this paper. PROSNOW is a project aiming at producing an operational climate service in order to

transfer it as a commercial service.

*Acknowledgements.* We thank our project partners and pilot ski resorts for many constructive discussions and providing data to improve the

manuscript. This project has received funding from the European Union's Horizon 2020 research and innovation programme under grant

agreement No. 730203.

**Appendix A:  Supplementary Material**

### A1    Further information on: Ski resort reference unit - SRU

In our approach, the spatial representations of ski areas and of the interpolated meteorological fields as well as the simulated

snowpack information differs in their saptial representation: the geometry type used for the ski slope is a vector-based polygon,

whereas the input and output of the snowpack models are based on a discrete approach, using regular grids of points. The

challenge for PROSNOW was to define an intermediary spatial object. This should be consistent with representations balanced

between the heterogeneity of meteorological conditions within ski slopes, the accuracy of snowpack models and the computa-

tion resources and data volume required for a daily update. Also it should take into account the localization of the snowmaking

facilities. The SRU, standing for Ski resort Reference Unit aims at fulfilling all these requirement. It can be end-user defined

including specific needs of the ski resort but it also can be processed automatically by chaining several operations. We stored

the vectorial GIS and attributes data of all nine pilot ski resorts in a PostgreSQL 10,7 DBMS and the crossing operations be-

tween rasters and vectors were performed with Python 2,7 with the GDAL/OGR along with NumPy. For automatic processing

of the SRUs we considered the following steps:

1. The association of each snowgun with a single ski slope is based on the spatial relation of the nearest neighbor to the ski
   slope.





2. The slope area covered by snowmaking is calculated by determination of the upper and lower altitude bounds for each snowgun. Considering that the mean surface covered by a single snowgun is approximately 1/3 ha, we applied a PLPG/SQL function to calculate the intersection between a slope and incremental 5 m buffer around the snowgun point until it reached at least 3,000 m$^2$. This buffer was then crossed with the combined ASTER-SRTM DEM v1.1 made available by the Copernicus European organization and we kept the buffer's altitude bounds which were then aggregated at the slope level to cover a continuous surface.

3. Once the snowtype attribute ("grooming only" or "with snowmaking") is defined for every slope, the according areas were then divided in smaller parts based on the elevation resolution. The initial DEM topography was re-classified with respect to the targeted SRU resolution. A value in the according numeric series was assigned to a pixel which value is in the range of the target value approximately half the resolution (i.e. for a 50 m resolution, the value 300 will be assigned at all DEM pixel which value is more or equal to 275 m and less than 325 m, and for a 300 m resolution between 150 m and 450 m. Contiguous pixels with the same values were merged and polygonized.

4. As small SRUs might occur by applying the above mentioned steps, e.g. at the beginning or ending of single pistes, we merged them in a post processing step with other small adjacent piste fragments having the same snowtype attribute. The final operation consisted in filling the missing snowtype attributes by calculating the average area value for each polygon from the DEM and its derivatives (slopes and aspects) and to match the output from snow models (all snowtype attributes value, average altitudes – and min/max altitude too – , slopes and aspects are also stored).



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



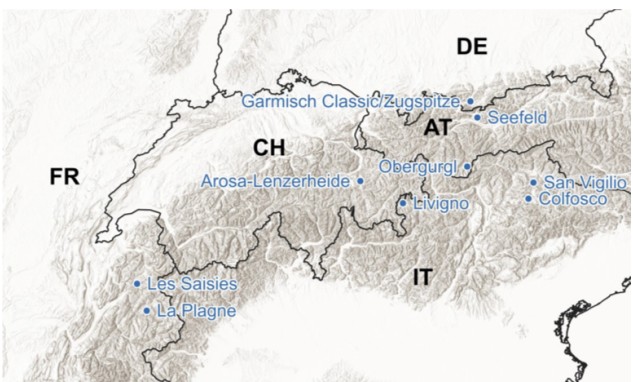

**Figure 1.** Locations of the PROSNOW pilot ski resorts.





**Figure 2.** An example of a ski resort disrectization into different SRUs elevation bands: 50, 100, 200, 300, and 400 meters. The figure shows the western part of Arosa-Lenzerheide. The different colors represent the different SRU areas.
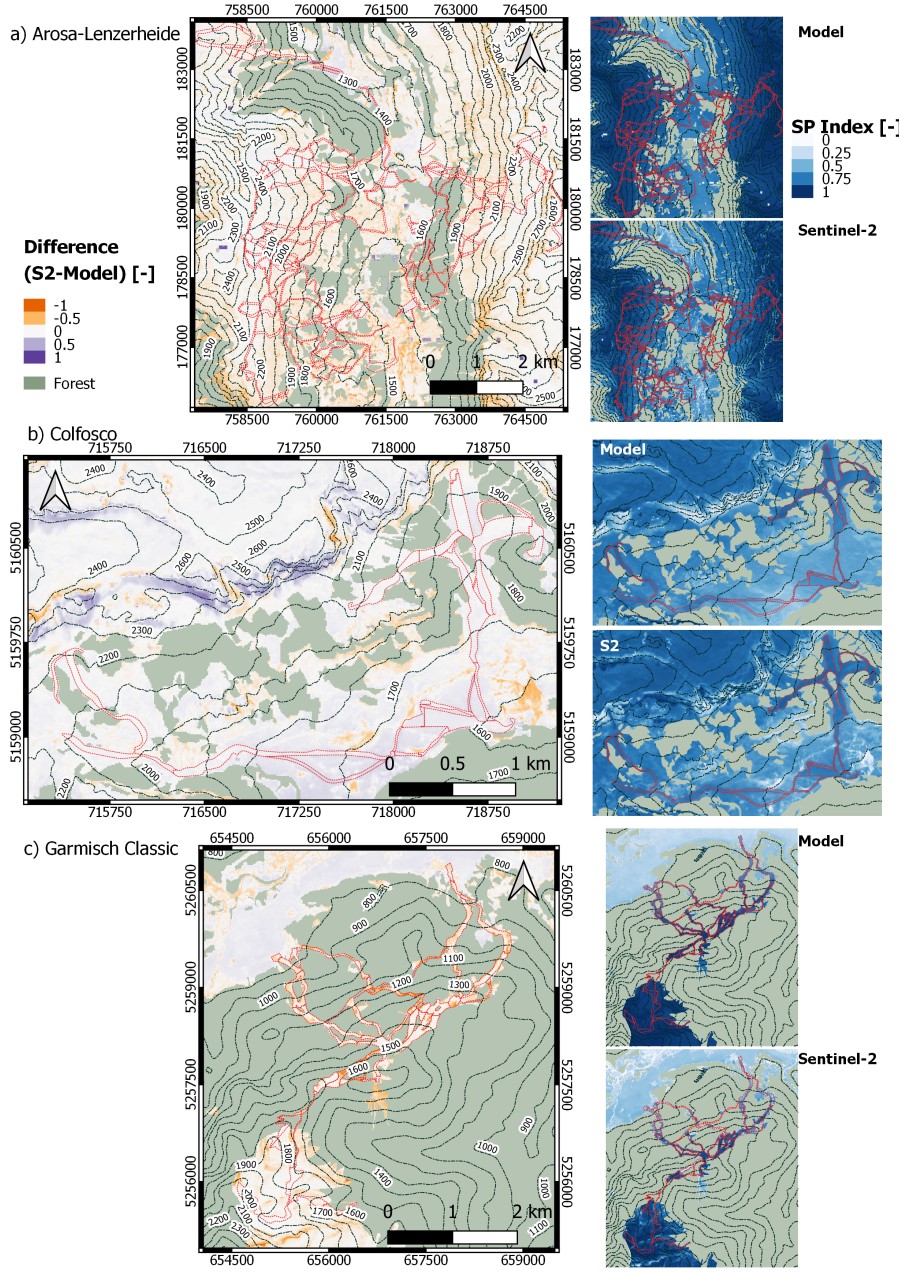

**Figure 3.** Snow persistence index difference between Sentinel-2 data and the model data (on the left) and SP indices for the simulation and Sentinel-2 (on the right) for each ski resort: a) Arosa-Lenzerheide, b) Colfosco, c) Garmisch Classic, d) La Plagne, e) Les Saisies, f) Livigno, g) Obergurgl, h) San Vigilio, and i) Seefeld. The period 2015-2020 was considered where valid Sentinel-2 data are available.
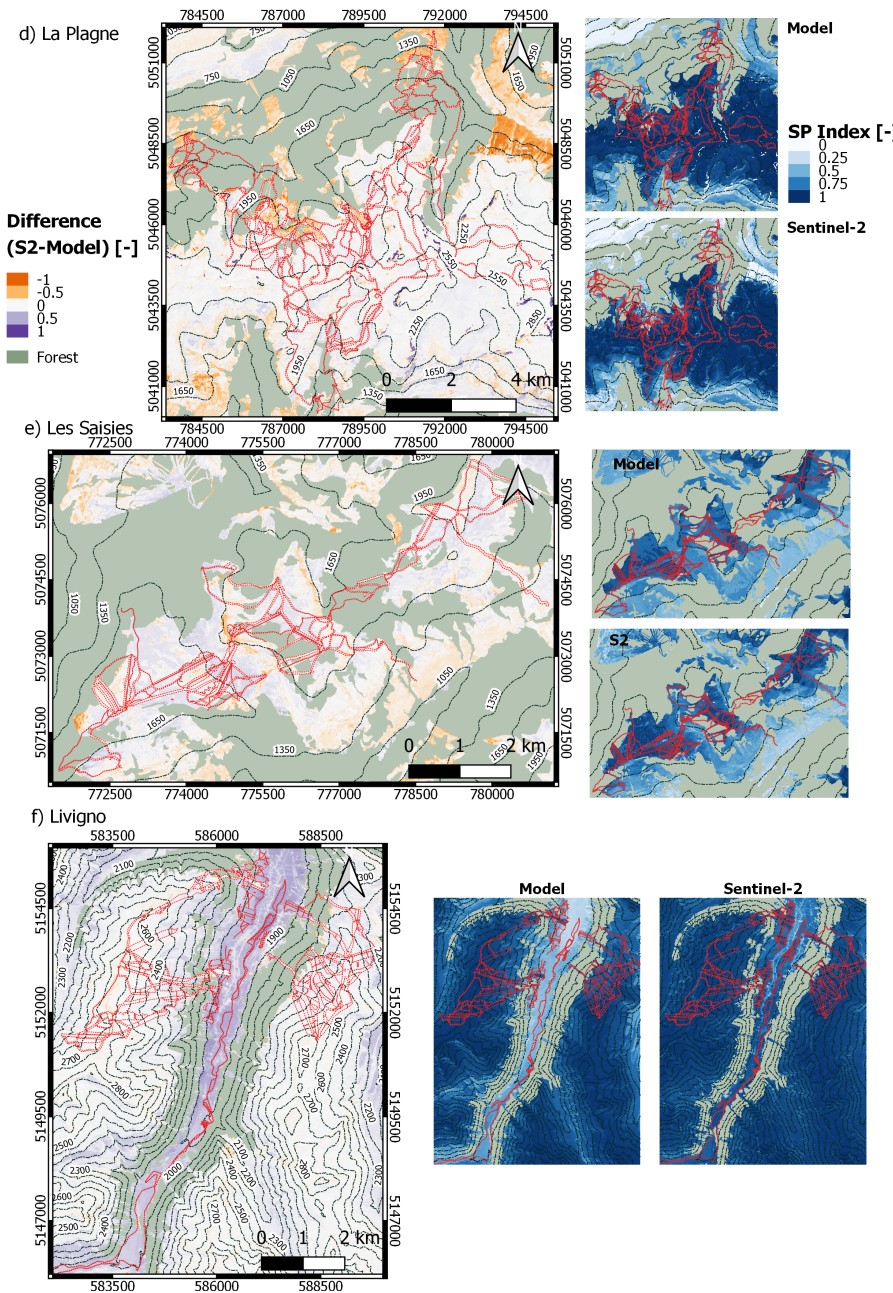

**Figure 3.** (Cont.) Snow persistence index difference between Sentinel-2 data and the model data (on the left) and SP indices for the simulation and Sentinel-2 (on the right) for each ski resort: a) Arosa-Lenzerheide, b) Colfosco, c) Garmisch Classic, d) La Plagne, e) Les Saisies, f) Livigno, g) Obergurgl, h) San Vigilio, and i) Seefeld. The period 2015-2020 was considered where valid Sentinel-2 data are available.

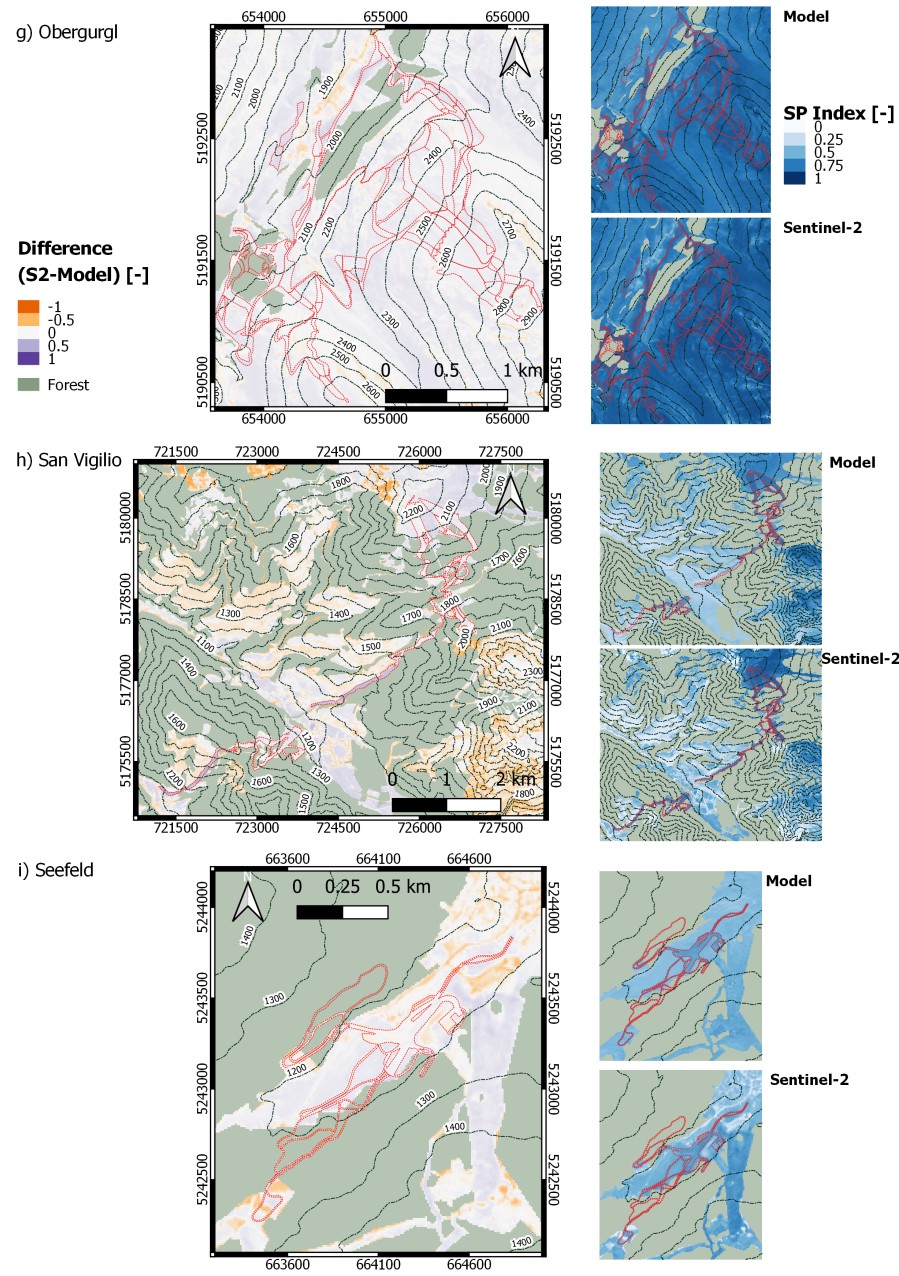

**Figure 3.** (Cont.) Snow persistence index difference between Sentinel-2 data and the model data (on the left) and SP indices for the simulation and Sentinel-2 (on the right) for each ski resort: a) Arosa-Lenzerheide, b) Colfosco, c) Garmisch Classic, d) La Plagne, e) Les Saisies, f) Livigno, g) Obergurgl, h) San Vigilio, and i) Seefeld. The period 2015-2020 was considered where valid Sentinel-2 data are available.





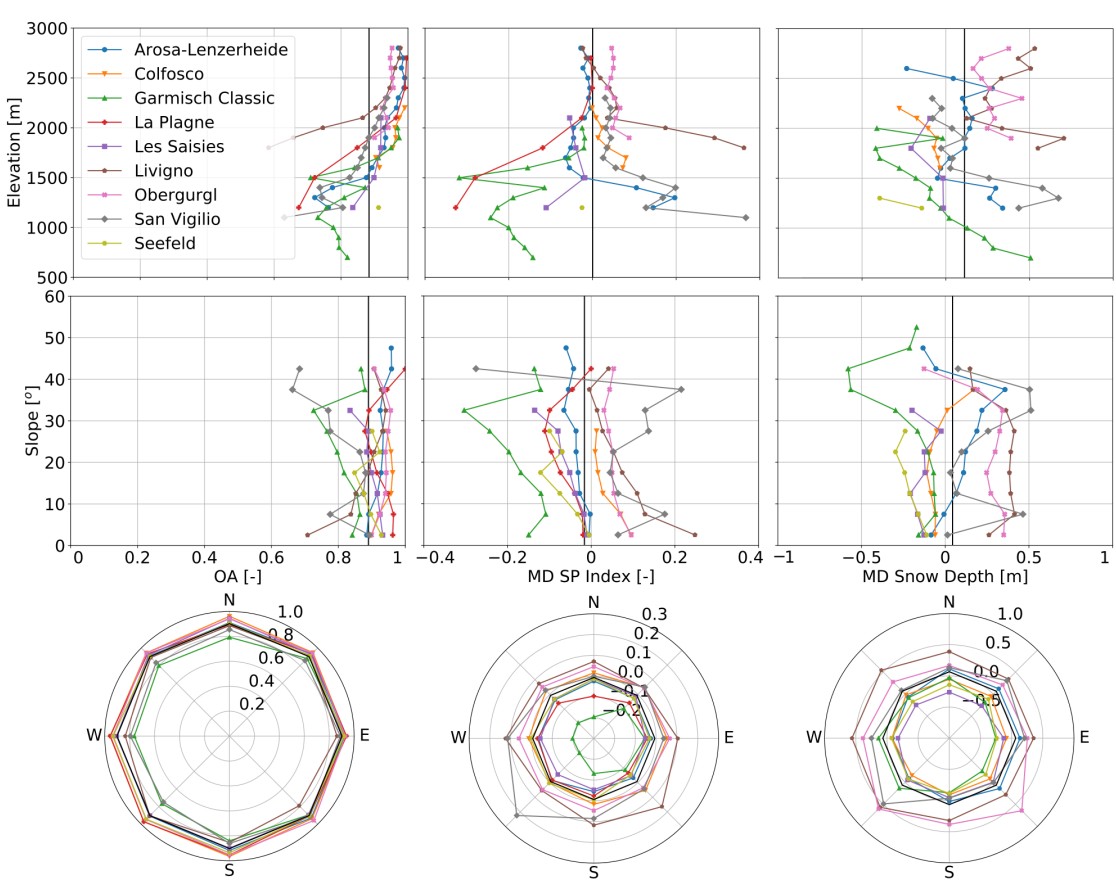

**Figure 4.** Comparison for 100 m elevation bands, 5 degrees slope bands and 45 degrees aspect bands for (left) overall accuracy between simulation and S2-data; (middle) mean deviation (MD) in snow persistence index between S2-data and simulation; and (right) mean deviation (MD) value between the simulated and GNSS-measured snow depth data. All metrics are computed for snow on the pistes only.



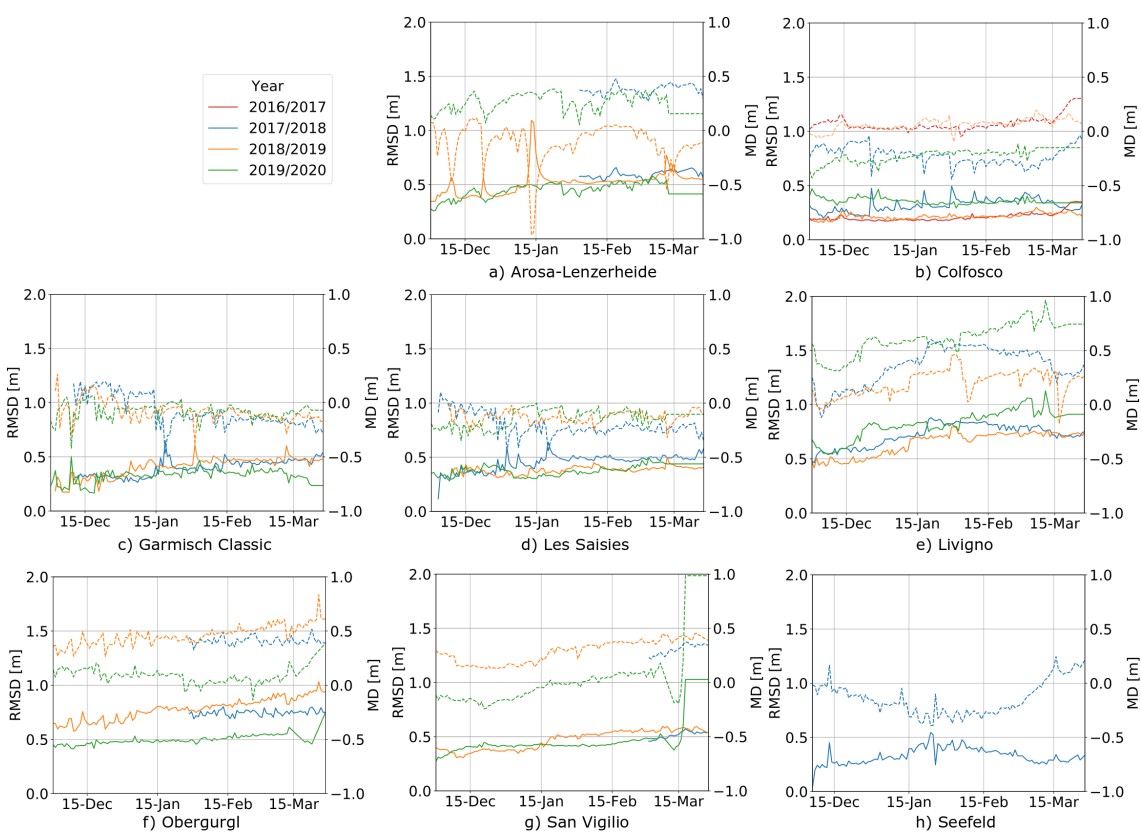

**Figure 5.** Root mean square deviation (RMSD) (continuous line) and mean deviation (MD) (dashed line) averaged over space between simulated and GNSS measured snow depth over time for the ski resorts. The period 2016-2020 was considered where valid GNSS measured snow depth data are available.

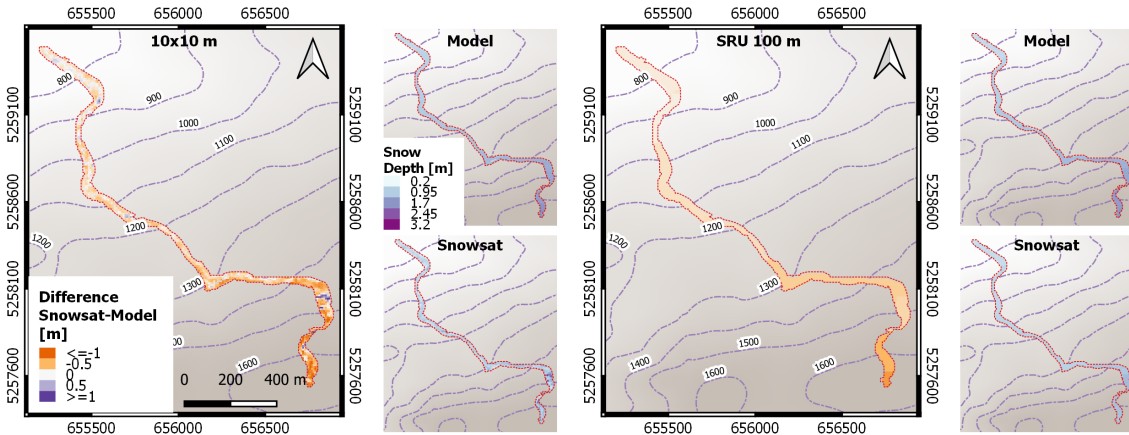

**Figure 6.** Example of spatial variability within a piste in Garmisch Classic (22.01.2018). On the left, RMSD represented as pixelwise difference between measured and modelled snow depth for a 10 m pixel resolution. The smaller figures represent the original modelled and measured snow depth. On the right, RMSD represented as averaged difference between measured and modelled snow depth for a 100 m band SRU resolution.



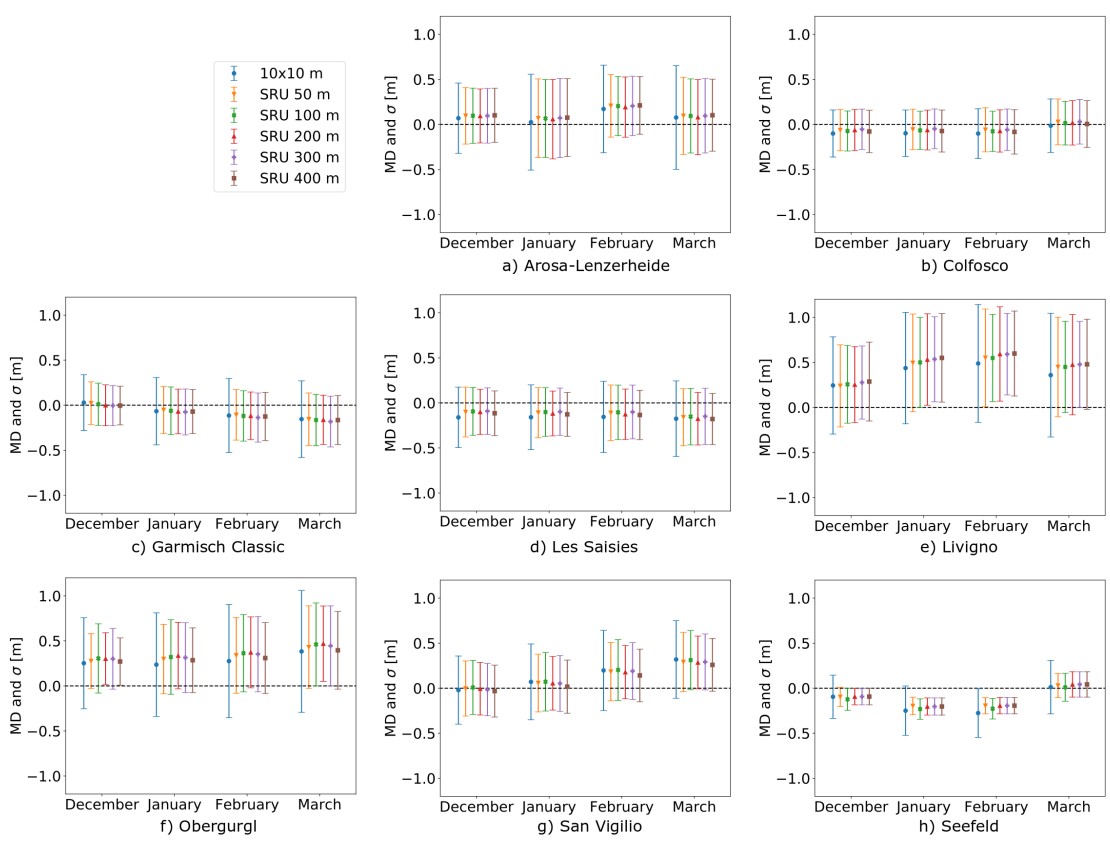

**Figure 7.** Overview of the global average and standard deviation between simulated and GNSS measured snow depth considering all the time steps and the different resolution of the SRUs. The data are analysed for the four months December, January, February and March.



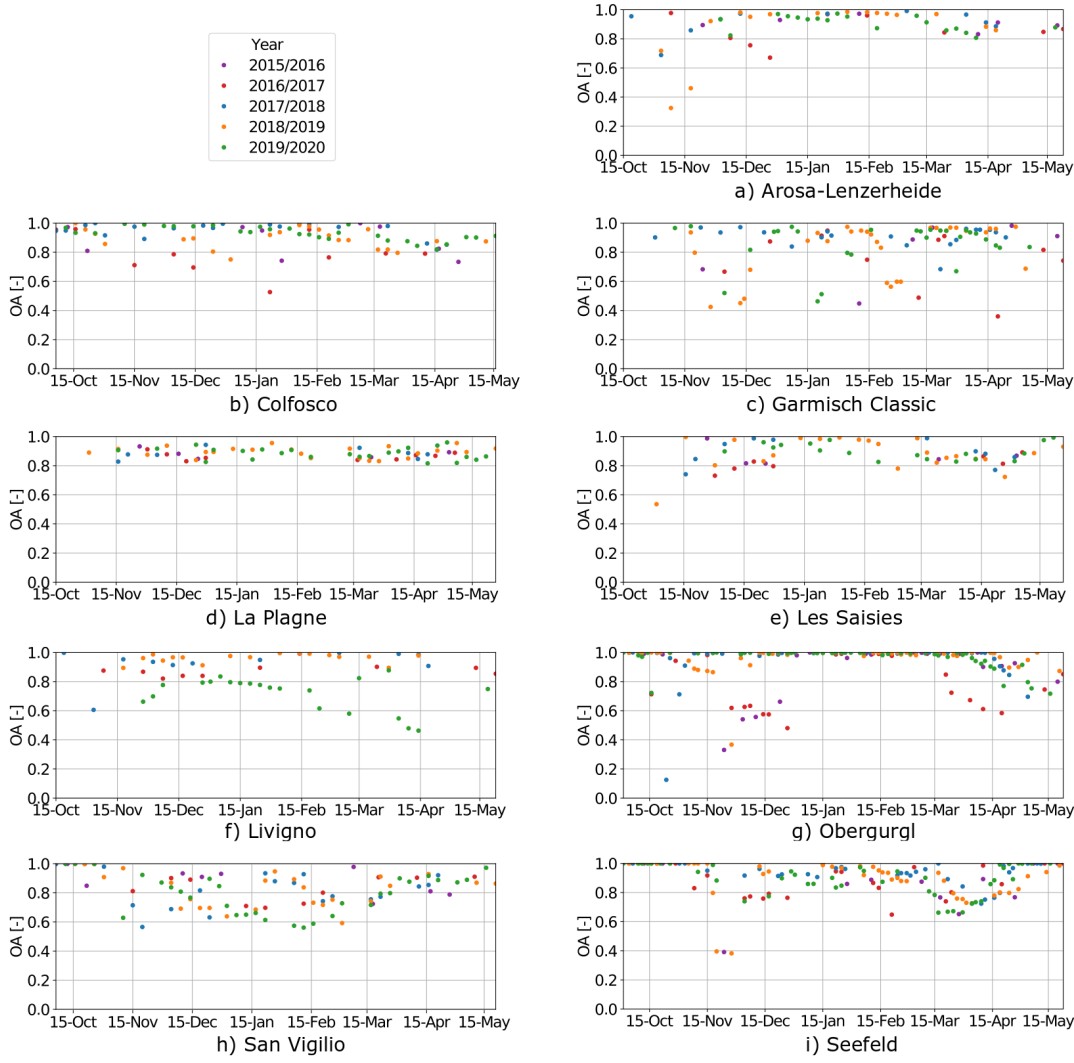

**Figure A1.** Overall agreement over time between simulation and S2-data for each ski resort for natural snow outside from the pistes. We considered images ranging from the winter season 2015/16 to the winter season 2019/20, except for Livigno where simulations are not available for the first season 2015/16 due to gaps in the meteorological forcing data.


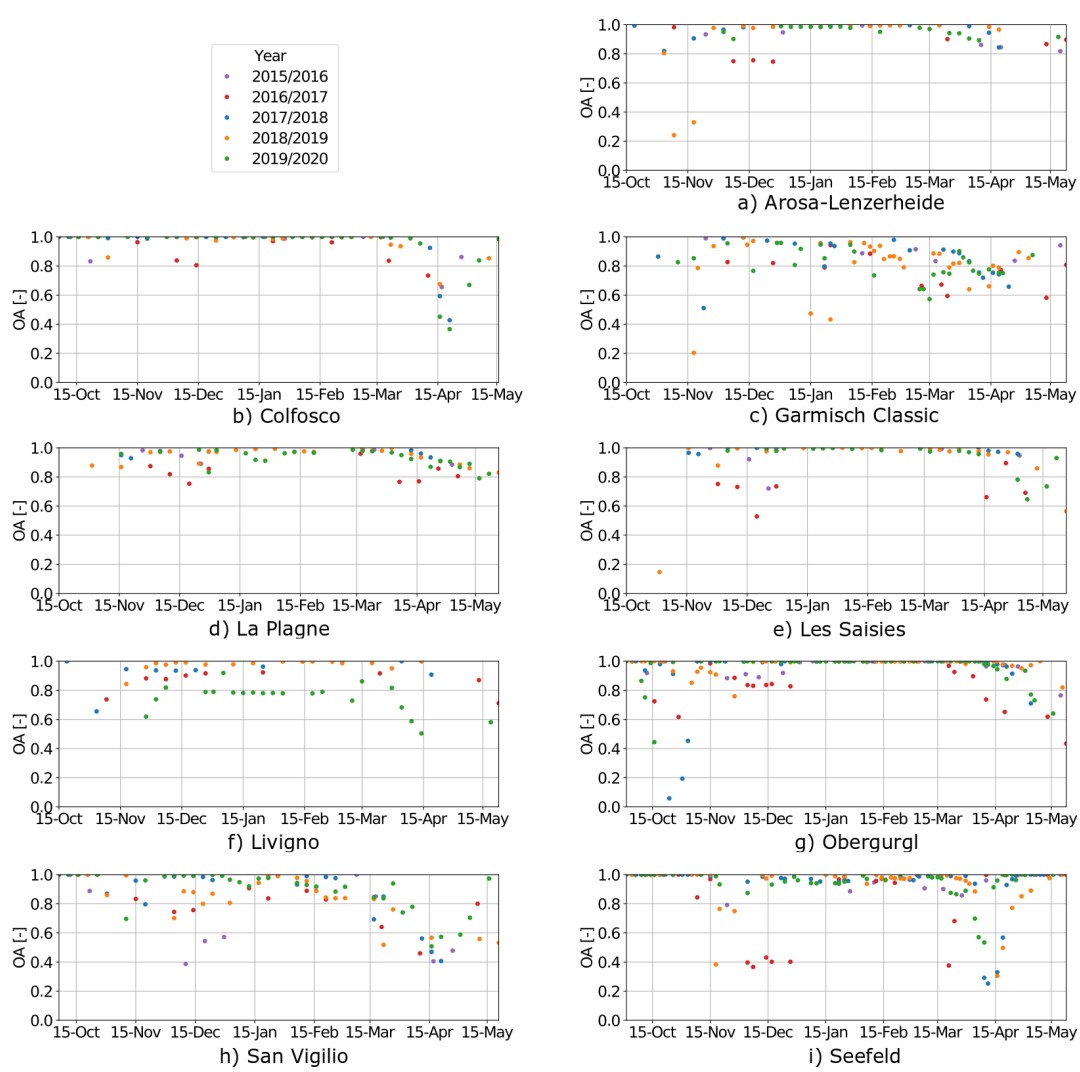

**Figure B1.** Overall accuracy trends over time between simulation and S2-data for each ski resort for machine made snow on the pistes. We considered images ranging from the winter season 2015/16 to the winter season 2019/20, except for Livigno where simulations are not available for the first season 2015/16 due to gaps in the meteorological forcing data.

**Figure C1.** Comparison for 100 m elevation bands, 5 degrees slope bands and 45 degrees aspect bands for (left) overall accuracy between simulation and S2-data; and (right) mean deviation (MD) SP index between simulation and S2-data. All the metrics are computed for natural snow only.





**Table 1.** Overview of the nine PROSNOW pilot ski resorts, the periode of available Sentinel-2 and measured GNSS snow depth data, and the used snow management configurations for the simulations based on the paper by Hanzer et al. (2020).

| Resort | Country | Elevation range (m a.s.l.) | Slope area (ha) | Sentinel-2 | GNSS | Configurations[a] |
|---|---|---|---|---|---|---|
| Arosa-Lenzerheide | CH | 1200-2865 | 384 | 2015-2020 | 2017-2020 | 2, 23, 31 |
| Colfosco | IT | 1531-2218 | 64 | 2015-2020 | 2016-2020 | 2, 23 |
| Garmisch Classic | DE | 708-2100 | 66 | 2015-2020 | 2017-2020 | 2, 7 |
| La Plagne | FR | 1250-3250 | 528 | 2015-2020 | - | 2, 11 |
| Les Saisies | FR | 1150-2069 | 214 | 2015-2020 | 2017-2020 | 2, 11 |
| Livigno | IT | 1816-2797 | 448 | 2015-2020 | 2017-2020 | 2, 23, 31 |
| Obergurgl | AT | 1930-2898 | 107 | 2015-2020 | 2017-2020 | 2, 23 |
| San Vigilio | IT | 1087-2274 | 119 | 2015-2020 | 2017-2020 | 2, 23 |
| Seefeld | AT | 1179-1251 | 79 | 2015-2020 | 2017-2018 | 2, 23 |

[a] (Hanzer et al., 2020)





**Table 2.** Example of the confusion matrix used in the results section. The analysis was split in three periods: beginning (B: October-November-December), middle (M: January-February) and end (E: March-April-May) of the season. TB: true positive; FP: false positive; FN: false negative; TN: true negative.

|  |  | Simulation | | |
|---|---|---|---|---|
|  |  | **Snow** | **Snow-Free** | **OA (%)** |
| **Sentinel-2** | **Snow** | TP [B; M; E] | FP [B; M; E] | OA (%) [B; M; E] |
|  | **Snow-Free** | FN [B; M; E] | TN [B; M; E] |  |





**Table 3.** Confusion matrix for all ski resorts referring to snow on the pistes. The values in the parentheses are referring to natural snow. The metrics in the square brackets are indicated also referring snow only to the beginning, middle and end of the season.

**Arosa-Lenzerheide**

| | | Snow | Snow-Free | OA (%) |
|---|---|---|---|---|
| Sentinel-2 | Snow | 82 (77) [71 (67); 99 (96); 76 (68)] | 3 (1) [2 (2); 0 (0); 5 (3)] | 92 (90) [85 (83); 99 (96); 93 (89)] |
| | Snow-Free | 5 (9) [13 (15); 1 (4); 2 (8)] | 10 (13) [14 (16); 0 (0); 17 (21)] | |

**Colfosco**

| | | Snow | Snow-Free | OA (%) |
|---|---|---|---|---|
| Sentinel-2 | Snow | 71 (63) [46 (41); 100 (89); 71 (63)] | 5 (5) [1 (5); 0 (4); 14 (8)] | 94 (91) [98 (93); 100 (92); 85 (87)] |
| | Snow-Free | 1 (4) [1 (2); 0 (4); 1 (5)] | 23 (28) [52 (52); 0 (3); 14 (24)] | |

**Garmisch Classic**

| | | Snow | Snow-Free | OA (%) |
|---|---|---|---|---|
| Sentinel-2 | Snow | 71 (45) [78 (44); 87 (73); 59 (29)] | 1 (6) [2 (16); 0 (8); 1 (2)] | 82 (84) [87 (77); 87 (83); 79 (87)] |
| | Snow-Free | 16 (10) [11 (7); 13 (9); 20 (11)] | 12 (39) [9 (33); 0 (10); 20 (58)] | |

**La Plagne**

| | | Snow | Snow-Free | OA (%) |
|---|---|---|---|---|
| Sentinel-2 | Snow | 89 (68) [87 (67); 97 (85); 88 (61)] | 1 (3) [1 (5); 0 (2); 0 (3)] | 93 (89) [92 (89); 97 (90); 93 (88)] |
| | Snow-Free | 6 (8) [7 (6); 3 (8); 7 (9)] | 4 (21) [5 (22); 0 (5); 5 (27)] | |

**Les Saisies**

| | | Snow | Snow-Free | OA (%) |
|---|---|---|---|---|
| Sentinel-2 | Snow | 83 (60) [76 (55); 100 (94); 81 (50)] | 3 (6) [6 (10); 0 (1); 1 (5)] | 91 (88) [88 (86); 100 (94); 91 (87)] |
| | Snow-Free | 6 (6) [7 (4); 0 (5); 8 (8)] | 8 (28) [11 (31); 0 (0); 10 (37)] | |

**Livigno**

| | | Snow | Snow-Free | OA (%) |
|---|---|---|---|---|
| Sentinel-2 | Snow | 81 (79) [78 (78); 88 (86); 79 (73)] | 11 (11) [8 (7); 12 (13); 14 (14)] | 87 (86) [88 (88); 88 (86); 85 (84)] |
| | Snow-Free | 2 (3) [4 (6); 0 (1); 1 (2)] | 6 (7) [10 (9); 0 (0); 6 (11)] | |
| | | **Snow** | **Snow-Free** | **OA (%)** |
| | | **Model** | | |



**Obergurgl**

| | | Snow | Snow-Free | OA (%) |
|---|---|---|---|---|
| **Sentinel-2** | **Snow** | 80 (72) [51 (42); 100 (99); 92 (84)] | 6 (6) [9 (10); 0 (0); 6 (6)] | 94 (93) [90 (88); 100 (99); 94 (92)] |
| | **Snow-Free** | 0 (1) [1 (2); 0 (1); 0 (2)] | 14 (21) [39 (46); 0 (0); 2 (8)] | |

**San Vigilio**

| | | Snow | Snow-Free | OA (%) |
|---|---|---|---|---|
| **Sentinel-2** | **Snow** | 61 (34) [48 (26); 93 (61); 47 (21)] | 12 (6) [4 (4); 5 (5); 30 (10)] | 85 (83) [90 (88); 93 (74); 69 (84)] |
| | **Snow-Free** | 3 (11) [6 (8); 2 (21); 1 (6)] | 24 (49) [42 (62); 0 (13); 22 (63)] | |

**Seefeld**

| | | Snow | Snow-Free | OA (%) |
|---|---|---|---|---|
| **Sentinel-2** | **Snow** | 51 (42) [33 (24); 97 (91); 42 (31)] | 3 (6) [0 (5); 0 (1); 8 (9)] | 91 (89) [92 (92); 97 (91); 87 (86)] |
| | **Snow-Free** | 6 (5) [8 (3); 3 (8); 5 (5)] | 40 (47) [59 (68); 0 (0); 45 (55)] | |

| | **Snow** | **Snow-Free** | **OA (%)** |
|---|---|---|---|

**Model**





**Table A1.** Total number of available Sentinel-2 data for each ski resort.

| Resort | Sentinel-2 |
| --- | --- |
| Arosa-Lenzerheide | 63 |
| Colfosco | 93 |
| Garmisch Classic | 97 |
| La Plagne | 72 |
| Les Saisies | 65 |
| Livigno | 62 |
| Obergurgl | 190 |
| San Vigilio | 104 |
| Seefeld | 164 |





**Table B1.** Effect of discretisation of the simulated results into different SRUs elevation bands: 50, 100, 200, 300, and 400 meters. The calculated root mean square error (RMSD), mean deviation (MD) and the standard deviation (STD) are between the GNSS measured and simulated snow depth. GNSS data were not available for La Plagne.

| Resort | SRU | Amount SRUs | Mean SRU size ($m^2$) | Slope average (°) | MD | RMSD | STD |
|---|---|---|---|---|---|---|---|
| Arosa-Lenzerheide | - | 38394 | 100 | 10.7 | 0.086 | 0.507 | 0.496 |
| | 50 | 533 | 7215 | 17.8 | 0.117 | 0.403 | 0.382 |
| | 100 | 312 | 12325 | 17.4 | 0.115 | 0.392 | 0.370 |
| | 200 | 215 | 17878 | 16.9 | 0.107 | 0.392 | 0.373 |
| | 300 | 175 | 21968 | 17.3 | 0.117 | 0.394 | 0.371 |
| | 400 | 151 | 25456 | 16.7 | 0.123 | 0.388 | 0.363 |
| Colfosco | - | 6378 | 100 | 12.5 | -0.079 | 0.287 | 0.273 |
| | 50 | 65 | 9832 | 14.3 | -0.038 | 0.244 | 0.238 |
| | 100 | 36 | 17731 | 13.7 | -0.050 | 0.234 | 0.225 |
| | 200 | 26 | 24551 | 13.5 | -0.045 | 0.233 | 0.240 |
| | 300 | 22 | 29015 | 13.3 | -0.033 | 0.236 | 0.231 |
| | 400 | 19 | 33597 | 14.0 | -0.057 | 0.253 | 0.244 |
| Garmisch Classic | - | 15874 | 100 | 16.4 | -0.077 | 0.380 | 0.393 |
| | 50 | 172 | 9443 | 18.6 | -0.074 | 0.284 | 0.268 |
| | 100 | 90 | 18046 | 18.3 | -0.084 | 0.285 | 0.266 |
| | 200 | 58 | 28003 | 17.9 | -0.088 | 0.273 | 0.253 |
| | 300 | 45 | 36093 | 16.3 | -0.100 | 0.282 | 0.258 |
| | 400 | 39 | 41645 | 16.0 | -0.092 | 0.271 | 0.250 |
| La Plagne | - | 52831 | 100 | 15.3 | - | - | - |
| | 50 | 994 | 5376 | 16.9 | - | - | - |
| | 100 | 644 | 8297 | 16.6 | - | - | - |
| | 200 | 437 | 12228 | 15.9 | - | - | - |
| | 300 | 366 | 14598 | 15.3 | - | - | - |
| | 400 | 322 | 16592 | 14.8 | - | - | - |
| Les Saisies | - | 17772 | 100 | 12.8 | -0.163 | 0.411 | 0.378 |
| | 50 | 487 | 3647 | 12.6 | -0.120 | 0.319 | 0.295 |
| | 100 | 316 | 5617 | 12.7 | -0.113 | 0.310 | 0.288 |
| | 200 | 240 | 7394 | 12.4 | -0.130 | 0.298 | 0.268 |
| | 300 | 213 | 8329 | 11.9 | -0.110 | 0.304 | 0.283 |
| | 400 | 213 | 8336 | 11.4 | -0.139 | 0.297 | 0.263 |



| Resort | SRU | Amount SRUs | Mean SRU size (m$^2$) | Slope average (°) | MD | RMSD | STD |
|---|---|---|---|---|---|---|---|
| Livigno | - | 47947 | 100 | 13.1 | 0.382 | 0.736 | 0.625 |
| | 50 | 493 | 9756 | 17.3 | 0.434 | 0.684 | 0.523 |
| | 100 | 308 | 15595 | 16.6 | 0.439 | 0.654 | 0.480 |
| | 200 | 216 | 22230 | 15.9 | 0.464 | 0.688 | 0.502 |
| | 300 | 164 | 29263 | 16.2 | 0.470 | 0.656 | 0.452 |
| | 400 | 143 | 33568 | 16.3 | 0.479 | 0.679 | 0.475 |
| Obergurgl | - | 10647 | 100 | 18.3 | 0.287 | 0.663 | 0.596 |
| | 50 | 163 | 6551 | 21.5 | 0.336 | 0.517 | 0.392 |
| | 100 | 102 | 10453 | 21.1 | 0.363 | 0.557 | 0.422 |
| | 200 | 66 | 16161 | 21.2 | 0.370 | 0.522 | 0.367 |
| | 300 | 58 | 18422 | 20.5 | 0.353 | 0.531 | 0.396 |
| | 400 | 48 | 22151 | 20.7 | 0.315 | 0.480 | 0.362 |
| San Vigilio | - | 12571 | 100 | 16.0 | 0.141 | 0.457 | 0.419 |
| | 50 | 121 | 10944 | 16.9 | 0.132 | 0.361 | 0.320 |
| | 100 | 78 | 16968 | 16.4 | 0.149 | 0.370 | 0.323 |
| | 200 | 50 | 26460 | 16.1 | 0.128 | 0.337 | 0.295 |
| | 300 | 44 | 30079 | 16.1 | 0.130 | 0.350 | 0.307 |
| | 400 | 49 | 27000 | 15.4 | 0.096 | 0.325 | 0.292 |
| Seefeld | - | 1427 | 100 | 5.8 | -0.152 | 0.328 | 0.270 |
| | 50 | 7 | 20537 | 10.4 | -0.115 | 0.179 | 0.107 |
| | 100 | 5 | 28918 | 9.8 | -0.143 | 0.210 | 0.126 |
| | 200 | 4 | 36147 | 8.3 | -0.112 | 0.179 | 0.105 |
| | 300 | 4 | 36147 | 8.3 | -0.112 | 0.179 | 0.105 |
| | 400 | 4 | 36147 | 8.3 | -0.112 | 0.179 | 0.105 |



**Table C1.** Inter-comparing the model simulations with Sentinel-2 data and GNSS measured snow depth for each ski resort.

| Resort | Sentinel-2 | GNSS |
|--------|-----------|------|
| Arosa-Lenzerheide | Slight overestimation of the snow line at the beginning of the season. | Ephemeral snowfall generates high disagreement. Increasing error throughout the season with a overestimation of the simulation. |
| Colfosco | Systematic error at the end of the seasons leading to an overestimation of the snow depth | Good agreement throughout the winter with slightly overestimation of the simulation. Especially at south-west exposed slopes encountered an overestimation of snow |
| Garmisch-Classic | Systematic errors at the beginning and end of the seasons lead to an overestimation of the snow coverage in these periods. However, it has to be noted that the snow season is shorter than in the other ski resorts and the area is very heterogeneous. | Good agreement throughout the winter with underestimation a the beginning and overestimation at end of the season of the simulations. Especially at north-east exposed slopes encountered an overestimation of snow in general. |
| La Plagne | Systematic error at the end of the seasons leading to an overestimation of the results. Especially at north exposed slopes encountered an overestimation of the snow depth. | Good agreement throughout the winter with slightly overestimation of the simulation. |
| Les Saisies | Systematic error at the end of the seasons leading to an overestimation of the snow depth. Especially at south exposed slopes encountered an overestimation of the snow. | No data available. |
| Livigno | Underestimation and overestimation of the snow at the beginning and end of the season. In addition, an underestimation of snow presence at the valley floor was encountered. | Good agreement at the beginning but increasing error during the season. Overestimation of the snow depth throughout the winter for the simulations. Especially at south-west exposed slopes encountered an overestimation of the snow |
| Obergurgl | Ephemeral snowfall misclassification generates high disagreement. | Good agreement at the beginning but increasing error during the season. Overestimation at the beginning and slightly underestimation of the snow depth at the end of the season for the simulations. Especially at south-west exposed slopes encountered an overestimation of the snow |
| San Vigilio | Systematic error at the end of the seasons leading to an underestimation of the snow depth. As it is a low altitude ski resort there are also some errors in the middle of the seasons. | Good agreement throughout the winter with slightly underestimation of the simulation in the middle of the season. |
| Seefeld | Systematic error at the beginning and end of the seasons leading to overestimation and underestimation of the snow depth. | Good agreement throughout the winter with slightly underestimation of the simulation in the middle of the season. |