# Peer review of "Evaluating a prediction system for snow management"

_The Cryosphere, 2021_

## Author Comment (AC2)

**RESPONSE TO RICHARD L.H. ESSERY**
**TO MANUSCRIPT tc-2021-56-RC1**

***Title:*** Evaluating a prediction system for snow management

**Authors:** Pirmin Philipp Ebner et al.

We thank Richard L.H. Essery for his positive feedback, constructive comments and suggestions. To your comments:

**Comment #1:** L9: Specifically, the comparison with Sentinel-2 data is for snow-covered area.

> *[ANSWER]* We will rephrase the sentence:
>> "*… more than 80 % for snow-covered area compared to the Sentinel-2 data .*"

**Comment #2:** L10: Redistribution of snow by skiers would not directly lead to a reduction in average snow depth. This may be a significant omission in simulations, but the statement in the abstract seems more confident than the discussion in the paper.

> *[ANSWER]* Maybe this sentence in the abstract was misleading. We actually meant that due to the redistribution of snow by skiers the variability of snow depth increases regarding the quite fine 10 m x 10 m resolution. This variability is of course visible in the GNSS snow depth measurements but are not simulated in the model. Therefore, we rephrased this sentence to:
>> *L10:* "*Potential sources for local differences of the snow depth between the simulations and the measurements are mainly due to the impact of snow redistribution by skiers or spontaneous local adaptions of the snow management, which were not reflected in the simulations.*"

**Comment #3:** L24: What is "early winter" in this context?

> *[ANSWER]* We refer "early winter" to October/November. We will change the sentence:
>> "*… to early winter (October/November) demand for perfect …*"

**Comment #4:** L29: What do these national percentages represent? Can a link be provided for Lalli et al. 2019?

> *[ANSWER]* The percentages represent the covered snow pistes with technical snow production. We will add a link for Lalli et al. 2019 and change the sentence to:
>
> > "Regarding pistes covered with snow originating from snow production, *Italy (90 %) …"*

**Comment #5:** L51: Monti et al. (2016) discusses initialization of a model with manual snow profiles, not remote sensing.

> *[ANSWER]* Correct, we will delete this reference in this context.

**Comment #6:** L69: This is the width of the elevation bands, not the elevation bands themselves.

> *[ANSWER]* Thanks for pointing this out. We will change the sentence:
>
> > "*… divided into a number of elevation bands with width ranging from 50 to 400 m."*

**Comment #7:** L88: The snow management configurations in Table 1 are incomprehensible without reading Hanzer et al. (2020). Brief descriptions and reasons for selecting them should be given to make this paper more self-sufficient.

> *[ANSWER]* We will change this sentence and add more information:
>
> > L88: *"… configurations for each ski resorts are shown in Table 1 and are selected to be comparable with the snow management configuration of each ski resort based on individual discussions with the ski resorts managers. In general, the basis for snow production is relying on resource saving assumptions, the features of the locally installed snow-making system as well as the opening and closing of each ski resort. The configurations where selected for each ski resort as follows and are described in more detail in Hanzer et al. (2020):*
> >
> > - *Configuration 2: No snow production; simulations are based on a natural snow only configuration, however with grooming activity*

- *Configuration 7: Snow production with a minimum required SWE of 150 kg m-2 using fans and a wet-bulb temperature of maximum -4°C*

- *Configuration 11: Snow production with a minimum required SWE of 150 kg $m^{-2}$ using lances and a wet-bulb temperature of maximum -4°C*

- *Configuration 23: Snow production with a minimum required SWE of 250 kg $m^{-2}$ using fans and a wet-bulb temperature of maximum -4°C*

- *Configuration 31: Snow production with a minimum required SWE of 250 kg $m^{-2}$ using lances and a wet-bulb temperature of maximum -6°C"*

**Comment #8:** L102: The SRU is a clever concept similar to the familiar HRU of hydrology, but it seems from the Supplementary Material that there is much more to the definition of SRUs than the slicing into elevation ranges described here.

*[ANSWER]* Correct but we decided not to include the whole SRU definition in the main paper. Therefore, we moved the detailed definition into the Supplementary Material part. However, we will add a reference to the Supplementary Material in the main text and will add the following sentence:

*L108: "Local snow managements play a major role in this as explained in more detail in the Supplementary Material A1."*

**Comment #9:** L158: In short, GNSS snow depth data were available for all pilot resorts except La Plagne.

*[ANSWER]* We will change this sentence accordingly.

**Comment #10:** L184: The 0-1 range of SP has already been stated.

*[ANSWER]* We will delete the sentence in Line 178-179.

**Comment #11:** L210: i = 0,…,N would be N+1 pixels

*[ANSWER]* Thanks, we will correct this.

**Comment #12:** L213: https://doi.org/10.1029/2010EO450004

> *[ANSWER]* We will change this sentence:
>
> > L213: "*A negative MD value indicates an overestimation and a positive MD value indicates an underestimation of the snow …*"

**Comment #13:** L226: Agreement between observations and models that the pistes are almost fully snow covered in the middle of the season is not surprising (these are ski resorts, after all!). A more interesting question, and a more important one for snow management, might be how much better the models perform in early and late season compared with simulations without snow management.

> *[ANSWER]* It is indeed more interesting to focus on the beginning (in particular) and the end of the season. We therefore emphasize these phases now a bit more but maintain the focus on the influence of snow management on the whole season. Our focus is on the complete ski resort with snow management. Even if we do not perform such an analysis, we can justify the goodness of the model without snow management based on our analysis for natural snow outside the pistes, mainly shown in Figure A1, B1 and C1 and in Table 3. The overall accuracy for natural snow is around 80 %. If we apply the snow management, we could further increase the overall accuracy. Based on this we can conclude that our models perform better also in early and late season compared with simulations without snow management.
>
> We will add the following sentence:
>
> > L226: "*The overall accuracy for natural snow is around 80 %. If we apply the snow management, we could further increase the overall accuracy. Based on this we can conclude that our models perform better, also in early and late season, compared with simulations without snow management.*"

**Comment #14:** L263: If slopes were not groomed, how are GNSS measurements available to quantify the model error? How does the lack of grooming lead to strong increases in RMSD?

> *[ANSWER]* GNSS measurements were available a day after this big snow fall event, therefore, we still could quantify the error. It is not the lack of grooming

but an overestimated snowfall in the model, which leads to an increase in RMSD (see Line 264). We will change the sentence to:

> *L263: "As a result, a large part of the ski area was closed and many slopes were no longer groomed at this date."*

**Comment #15:** L278-279: Is something missing from this sentence? It does not seem to make sense.

> *[ANSWER]* We will change the sentence to:
>
> > *Line278: "... in coarser clusters tends to mask the variability in the error in terms of ..."*

**Comment #16:** L281: Figure 7 is referred to before Figure 6

> *[ANSWER]* Thanks for pointing this out, we will change Figure 7 and 6.

**Comment #17:** Figure 3: It seems counterintuitive that the brightest colour is the lowest snow persistence.

> *[ANSWER]* We would like to keep it as it is as we would like to assign the colour white to '0' regarding the SPI as we did assign 0 also to the difference, although it might be somehow counterintuitive.

**Comment #18:** Figure 4: Having a zero line that is not in the centre of a radar plot is confusing. Absolute errors might be better, or at least highlight the zero line.

> *[ANSWER]* We will change Figure 4 and highlight the zero line (same for Figure C1).

**Comment #19:** Figure 5: Having RMSD and MD on the same figure but with different axes is very confusing and makes it difficult to tell at a glance if an error is small or large. Using a single axis would compact the error range but would be much clearer (this is common in evaluations of weather forecast errors)

> *[ANSWER]* We will change Figure 5 as shown below. In our opinion, the separation in solid lines for RMSD and dashed lines for MD should be now clear enough by adding more information for clarification also in the figure subtitle. Moreover, we included information on simulated and measured SD below, as

we believe this helps to better interpret MD and RMSD in the course of time over the season with varying SD.

[Figure]

Figure 5: Root mean square deviation (RMSD) (upper subplots: solid line, left axis) and mean deviation (MD) (upper subplots: dashed line, right axis) averaged over space between GNSS measured snow depth (SD) (lower subplots: solid line, left axis) and simulated SD (lower subplots: dashed line, right axis) over time for the ski resorts. Within the period 2016-2020 we considered all valid GNSS measured snow depth data which were available.

**Comment #20:** Figure 6: What is the nature of the large measured snow depth between 1400 m elevation? Is there a bump in the snow surface or a dip in the ground surface that is not resolved by the 10 m model?

*[ANSWER]* Indeed, there is a bump in the snow depth measured by the GNSS. It is due to a dip in the ground surface which was filled with snow to level the piste leading to a higher measured snow depth.

We will add the following sentence in the caption of figure 6:

Figure 6: "*The nature of the large measured snow depth by GNSS at around 1400 m elevation is due to a dip in the ground surface which was filled with snow to level the piste. This led to a higher GNSS measured snow depth compared to the simulations.*"

**Comment #21:** Figure 7: Little variation is seen in MD between resolutions. Text in 4.3 discusses reduction in RMSD, so that might be a better variable to show.

*[ANSWER]* Thanks for pointing this out. We will add information on RMSD and change Figure 7 to:

[Figure]

Figure 7: Overview of the root mean square deviation (RMSD, hollow symbols), global average (MD, filled symbols) and standard deviation (σ, extend of error bars) between simulated and GNSS measured snow depth considering all the time steps and displayed for each SRU resolution. The presented data are analysed for the four months December, January, February and March where GNSS data were available.

**Comment #22:** Table C1: Why does the column for Sentinel-2 contain statements about errors in snow depth?

*[ANSWER]* Thanks for pointing this out. It is not "snow depth" but "snow covered area" – we will change this.

The authors

---

## Author Comment (AC3)

**RESPONSE TO P.A.B. Bartlett**

**TO MANUSCRIPT tc-2021-56-RC2**

***Title:***      Evaluating a prediction system for snow management

**Authors:**   Pirmin Philipp Ebner et al.

We thank P.A.B. Bartlett for his positive feedback, constructive comments and suggestions. To your comments:

**Comment #1:** Line 32-34: Have any authors or ski resorts conducted a cost-benefit analysis for making snow that melts over various periods of time? Early season production of artificial snow may very well melt, leading to loss of the snow-cover, but this is likely weighed against income derived by the resort being able to remain open for a period of time. I suspect many resorts have some criteria and that a decision to make artificial snow is based on the likelihood that it will last long enough to recoup the cost. I don't expect these questions to be answered in this paper, but I wonder if examples of such information exist such that they might inform the discussion.

    *[ANSWER]* We found some information to answer this question in a study of Köberl et al. (2021) which is under review but should be a accepted soon. We will add the following sentences:

        Line 34: "Based on a study by Köberl et al. (2021, under review) the "uncertainty surcharge" of snow produced due to imperfect knowledge about upcoming weather and snow conditions paired with high risk aversion is likely to represent a noticeable share of total snow production and related water consumption as well as of total snow management operating costs. Depending on the pilot ski resort, respondents expect that perfect knowledge would reduce the amount of technical snow needed by 10% to 45%, the amount of water needed by 10% to 40%, and total snow management operating costs by 5% to 20%. Hence, there seems to be room for services that are able to improve the ski resorts' current ability to anticipate weather and snow conditions."

    We will add this in the revised manuscript.

        Köberl, J, François, H., Cognard, J., Carmagnola, C., Prettenthaler, F., Damm, A., and Morin. S.: The demand side of climate services for realtime snow management in Alpine ski resorts: some empirical insights and implications for climate services development, Climate Services, under review, 2021.

**Comment #2:** Line 64: Change "snow-covered maps" to "snow-cover maps".
*[ANSWER]* We will change it.

**Comment #3:** Line 67: Change "unit" to "units".
*[ANSWER]* We will change it.

**Comment #4:** Line 88: Change "The used snow management configurations for" to "The snow management configurations employed for".
*[ANSWER]* We will change it.

**Comment #5:** Line 106: Aggregating as a post-processing step simplifies the presentation but not the computation. Has it been tested whether similar results are obtainable employing these aggregated areas for the simulations?
*[ANSWER]* No, we didn't perform simulations on the aggregated areas since this would require a completely new model setup. As the Sentinel-2 and the GNSS snow depth data are given as rasterized data, we decided to perform the simulations similar and aggregate the simulated snow depth in a post-processing step. Another reason is that we would not be able to capture a realistic shape of the pistes, which are often quite narrow (approx. 10 m), with a coarser resolution as with Amundsen and Alpine3D we use models which rely on raster cells for calculation. Therefore, in our opinion it makes more sense to aggregate the 10 m x 10 m raster cells alongside the 'real' piste shapes with the altitudinal band concept we chose for aggregation.

**Comment #6:** Line 152: Change "This technique relies on differential GNSS signals and takes measurements without snow depth on the slopes as a reference into account." To "This technique relies on differential GNSS signals, comparing the snow-free (i.e. zero snow depth) reference signal with those obtained during the snow season, to obtain snow depth.
*[ANSWER]* We will change it.

**Comment #7:** Line 156: Snowsat and Leica are not defined.

*[ANSWER]* Snowsat and Leica are companies providing GNSS snow-depth measurements from the grooming machine. We will rephrase the sentence:

> *"… data were provided by the companies SNOWsat and Leica Geosystems AG and were …"*

**Comment #8:** Line 169: Change "constrains" to "constraints".

*[ANSWER]* We will change it.

**Comment #9:** Line 187: Change "Additionally" to "In addition".

*[ANSWER]* We will change it.

**Comment #10:** Figure 4: Are the bold vertical lines in each plot of Figure 4 the intra-model means? This should be defined in the graph pane or in the caption.

*[ANSWER]* The bold vertical lines are the intra-model means. We will add this in the caption..

**Comment #11:** Line 278: Is this averaging effect desirable? Instead of "allows minimizing the error", would "tends to mask the error" be a more accurate description of what is happening? Later, it is discussed that there may be a benefit to this, but I would still use "mask" perhaps as "tends to mask the variability in the error".

*[ANSWER]* Thanks for this good suggestion. Indeed, we don't want to "minimize the error" but to find out what is happening. We will change the sentence to:

> *"… in coarser clusters tends to mask the variability in the error in …"*

**Comment #12:** Why is Figure 7 presented before Figure 6? I would rename the figures.

*[ANSWER]* Thanks for pointing this out, we will change the order of Figure 7 and 6.

**Comment #13:** Line 309: Remove "it".

*[ANSWER]* We will remove it.

**Comment #14:** Line 315: Change "use" to "uses".

*[ANSWER]* We will change it.

**Comment #15:** Line 321: Change "in average" to "on average".

*[ANSWER]* We will change it.

**Comment #16:** Line 325: Change "e.g., rapid snow melt inside the catchment are hardly to be matched correctly by the models" to "e.g., the ensuing rapid snow melt inside the catchment is difficult to simulate accurately".

*[ANSWER]* We will change it.

**Comment #17:** Line 337-338: This is not a sentence.

*[ANSWER]* We will change the sentence:

> "*The GNSS data can only be used as ground observation with some restrictions.*"

**Comment #18:** Line 352-354: The authors should define some accuracy requirements for the snow models to meet the needs of the ski resorts.

*[ANSWER]* Unfortunately, we can't really add some accuracy requirement in this case. For the ski resort it is more important that they can reach at least the minimum snow depth (with a certain probability!?) before season opening and hold it. But we can say that the accuracy of all three snow models is generally sufficient (see Hanzer et al. (2020)) because the uncertainty coming from the meteorological/climatological input is much larger.

**Comment #19:** Line 357: I would not include the errors in the S and SW facing pistes with snow redistribution. These errors are caused by more rapid ablation because these pistes are exposed to high solar radiation during the warmest part of the day. I wouldn't classify that as redistribution although both are important.

*[ANSWER]* We will change this point to:

> "*… (2) snow redistribution by the groomers;*"

and we will add an additional point:

> "... (3) rapid ablation (e.g. south, south-west exposed pistes) due to high solar radiation;"

**Comment #20:** Line 393-394: I suspect that the ski resorts would know the minimum snow depth required. The research should attempt to determine whether the models can simulate snow depth with sufficient accuracy to enable the resort managers to maintain the optimum and minimum viable snow depth in a more efficient way.

> *[ANSWER]* This is correct and currently an additional publication is in preparation investigating this question for specific ski areas. Detailed studies for each ski resort are needed but this was not within the scope of this paper. We will add the following sentence and include a reference to the paper:
>
> > Line 397: "*However, further studies to determine whether the models can simulate snow depth with sufficient accuracy to enable the resort mangers to maintain the optimum and minimum viable snow depth in a more efficient way, are needed and will be attempted in the future (Köberl et al. 2021).*"
> >
> > > *Köberl, J, François, H., Cognard, J., Carmagnola, C., Prettenthaler, F., Damm, A., and Morin. S.: The demand side of climate services for real-time snow management in Alpine ski resorts: some empirical insights and implications for climate services development, Climate Services, under review, 2021.*

**Comment #21:** Overestimation of snow depth and S and SW facing pistes could be addressed by having the incoming radiation adjusted for slope and aspect. I am not asking for this to be done in this paper, but it would be an obvious improvement for the next paper.

> *[ANSWER]* In all models the adjustment of the incoming radiation for slope and aspect is already implemented. It seems like that the snow redistribution of the skier plays a major role but further validations on ski pistes are needed.

The authors

---

## Author Comment (AC4)

**RESPONSE TO ANONYMOUS REFEREE**
**TO MANUSCRIPT tc-2021-56-RC3**

***Title:*** Evaluating a prediction system for snow management

**Authors:** Pirmin Philipp Ebner et al.

We thank the anonymous referee for his positive feedback, constructive comments and suggestions. To your comments:

**Comment #1:** L153-155 Is there any reference to show the accuracy of GNSS? Also, does the GNSS has similar accuracy for wet snow?

> *[ANSWER]* We used the information provided by the companies' webpage (Leica and SNOWsat) to define the accuracy. It makes no difference in snow accuracy if the snow is dry or wet. In general, this technique is based on differential GNSS measurements, which allow an accuracy of snow depth measurements of a few centimetres. During the snow-covered period, the relative position of the groomer is tracked at each location. In combination with a precise digital elevation model derived in snow-free conditions (reference), the snow depth at a certain date and location is measured. The value that results after deducting the vehicle height is then compared with the altitude of a digital terrain model without snow cover stored in the system. The snow depth at the current vehicle position is the difference between these two altitudes.

> We will add the following sentence:
> > Line 155: "*GNSS snow depth measurements were provided by the companies Leica-Geosystems and SNOWsat.*"

**Comment #2:** L241-242 Can you point out where the biggest difference due to snow gliding or avalanche in Figure 3? Also, this discrepancy may be reduced by integrating avalanche dynamics model. Do you have a plan to integrate a snow redistribution model and avalanche dynamics model into this system? If there are any views for future implementation of them, description of it is desirable.

*[ANSWER]* At current stage, we do not plan to implement this in the models, however, we agree it could be an interesting topic for further research.

**Comment #3:** L250-254, Figure 4:   I guess that the better accuracy in high altitude is due to the ratio of snow cover area is near 1 (it may be most of them are true positive). Including the figure of simulated or observed snow cover ratio for each elevation and slope direction helps the relation of this ratio with OA.

*[ANSWER]* There was a mistake. We don't mean Figure 3 but Figure 4 (left). We will correct this and will also add the following sentence:

L249: *„A better accuracy is obtained in high altitudes due to the fact that the ratio of snow cover area is near 1."*

**Comment #4:** L255-256 Figure 5 shows the amount of MD and RMSD for snow depth. I think the information snow depth is also necessary to check relative errors. Can you add the figure of snow depth data for simulation and observation?

*[ANSWER]* Yes, we will include a sub-plot of snow depth for simulation and observation beneath the MD and RMSD sub-plots for each resort in Figure 5. We will change Figure 5 as shown below. In our opinion, the separation in solid lines for RMSD and dashed lines for MD should be now clear enough by adding more information for clarification also in the figure subtitle. Moreover, we included information on simulated and measured SD below, as we believe this helps to better interpret MD and RMSD in the course of time over the season with varying SD.

[Figure]

Figure 5: Root mean square deviation (RMSD) (upper subplots: solid line, left axis) and mean deviation (MD) (upper subplots: dashed line, right axis) averaged over space between GNSS measured snow depth (SD) (lower subplots: solid line, left axis) and simulated SD (lower subplots: dashed line, right axis) over time for the ski resorts. Within the period 2016-2020 we considered all valid GNSS measured snow depth data which were available.

**Comment #5:** L262-263 Although I haven't used and am not familiar with the grooming module, this error seems to be reduced if this module can turn on and off depending on the situation. This result can make suggestions to add them to improve the system.

> *[ANSWER]* Correct but the snow management configurations of the simulations are currently not adapted to the daily snow management decision of the ski resorts managers. However, in future, we aim to include this; it is ongoing work.

**Comment #6:** L278-280 I think the averaging effects for RMSD can be avoided when 10m meshed GNSS (not averaged) and SRU averaged simulated data are compared. In this case, larger SRU size leads to larger RMSD. This comparison is not a requirement, but it is worth a try.

> *[ANSWER]* In general, we think that this makes not so much sense to test this as the spatial variability of the 1 m resolution of the original GNSS data is too high to get plausible results compared to the simulated data. We decided to calculate the errors of the averaged snow depths, where the average is calculated with respect to the different SRU discretizations. This is of course (according to the reviewer) introducing averaging effects. Regarding the updated Figure 7 (see below), RMSD is getting smaller for coarser resolutions, whereas MD is more or less stable with variations that are not systematic. The RMSD shows that coarser resolutions work better due to these averaging effects, and this is what we actually want to show with this analysis. In other words, the simulations do not capture the high spatial variability of the snow depth as already mentioned in the manuscript. The aim was to find a good trade-off between high variability of GNSS and the inevitable coarse resolution of the SRU. According to this figure we state that 50 or 100 m altitudinal bands are a good trade-off in this sense. The potential analysis mentioned by the reviewer would only be another point of view for getting the same result.

[Figure]

Figure 7: Overview of the root mean square deviation (RMSD, symbols), global average (MD) and standard deviation (σ, error bars) between simulated and GNSS measured snow depth considering all the time steps and the different resolution of the SRUs. The data are analysed for the four months December, January, February and March where GNSS data were available.

**Comment #7:** L355-362 I think it would be more informative if there is some mention of future plan, actuality to achieve, and level of importance for the improvement to resolve (1) - (5).

*[ANSWER]* We agree and will add the following paragraph to the "Conclusions and Outlook" section:

> Line 423: "*Additionally, a detailed analysis to show the accuracy of the GNSS system to measure the snow depth is needed to validate the system. Moreover, integrating a snow redistribution model and an avalanche dynamics model into this system would help to point out where the biggest differences due to snow gliding or avalanches is given*

*between the Sentinel-2 data and the simulations. Further studies on the topographic complexity of the snow-free terrain and the rather smooth piste surface are needed to e.g. implement an index of surface smoothing compared to the bare ground. Future studies investigatinghow skiers redistribute snow under certain meteorological conditions in combination with topographic conditions (e.g. aspect, slope angle…) would also help to overcome further potential errors."*

The authors

---

## Author Response (AR2)

**MANUSCRIPT tc-2021-56**

***Title:***      Evaluating a prediction system for snow management

**Authors:**   Pirmin Philipp Ebner et al.

Dear Mr. Niwano,

We want to apologize for this inconvenience, which has brought with the rebuttal letter. This will not happen again. Thank you for the notice.

Below you can find our answers to the comments. The lines correspond to the latest track changed manuscript.

Kind regards

The authors

**Comment #1:** Equation (2) has a new error: the summation range should be either i = 0 to N-1 or i = 1 to N.:

    *[ANSWER]* We changed it to: i = 1 to N.

**Comment #2:** Defining MD positive for model overestimation would be more conventional but not essential.

    *[ANSWER]* We decided to keep it as it is but, in the future, we will do it the more conventional way

**Comment #3:** L. 19: "in future" -> "in the future"

    *[ANSWER]* We changed it accordingly.

**Comment #4:** L. 41: "the pilot ski resort" -> "ski resorts"? Is the word "pilot" necessary here? Please check this point again.

    *[ANSWER]* We changed "pilot ski resort" to "ski resort" in the whole manuscript.

**Comment #5:** L. 41: "respondents": Intention of this word is unclear. Please specify more in detail.

    *[ANSWER]* We meant with "respondents" the operating managers but agree that this might have been unclear. Therefore, we changed it to "operating managers".

**Comment #6:** L. 262: "because of the overestimation of the ablation process in the snow model" -> "because of the overestimation caused mainly by the ablation process in the snow model"

    *[ANSWER]* We changed it accordingly.

**Comment #7:** L. 374 ~ 375: Suggest rephrasing "The GNSS data can only be used as ground observation with some restrictions. There are several problems which might affect the quality of the data ~" -> "There are several problems which might affect the quality of the GNSS data ~"

    *[ANSWER]* We changed it accordingly.

**Comment #8:** L. 436: "in a more efficient way, are needed and will be attempted" -> "in a more efficient way are needed, and will be attempted"

[ANSWER] We changed it accordingly.

**Comment #9:** Figure 6 caption: You can remove the word "global average", because it is not used in the running text and the figure. Also consider again the comment #5 in CC1.

[ANSWER] We removed 'global average' and changed the caption to:

"Overview of the root mean square deviation (RMSD, hollow symbols mean deviation (MD, filled symbols) and standard deviation ($\sigma$, extent of error bars) between simulated and GNSS measured snow depth considering all the time steps and displayed for each different SRU resolution for the entire ski resort. The presented data are analysed for the four months December, January, February and March where GNSS data were available."

The authors